# What controls the coarse sediment yield to a Mediterranean delta The case of the Llobregat river (NE Iberian Peninsula)

Juan P. Martín-Vide, Arnau Prats-Puntí, Carles Ferrer-Boix*

Technical University of Catalonia, *Serra Húnter fellow

Jordi Girona 1-3, D1, 08034 Barcelona, Spain

*Correspondence to*: Juan P. Martín-Vide (juan.pedro.martin@upc.edu)

**Abstract.** The human pressure upon an alluvial river in the Mediterranean region has changed its riverine and deltaic landscapes. The river has been channelized in the last 70 years while the delta is being retreating for more than a century (a set of data unknown, so far). The paper concentrates on the fluvial component, trying to connect it to the delta evolution. Is the channelization responsible for the delta retreat? We develop a method to compute the actual bed load transport with real information of the past river morphology. The paper compares the computation with very limited measurements, among which bulk volumes of trapped material at a modern, deep river mouth. The decrease in sediment availability in the last 30 km of the channelized river is deemed responsible for the decrease in the sediment yield to the delta. Moreover, power development and flood frequency should be responsible for a baseline delta retreat along the 19 century. The sediment trapping efficiency of dams is less important than the flow regulation by dams, in the annual sediment yield. Therefore, it is more effective a step back from channelization than to pass sediment at dams, to provide sand to the beaches.

## 1 Introduction and objective

The framework for this research is the mankind pressure upon an alluvial river in the Mediterranean region. The paper aims at showing how and why the riverine and deltaic landscapes have changed. The time frame of the research is the last 70 years, over which the main pressure has been one of river channelization, yet some information prior to this period is necessary to understand the long term trends. The practice of channelizing a river generally involves increasing channel capacity and so, an erosional response is to be feared, although this is not always the case (Simon and Rinaldi, 2006). Typically, it also involves narrowing of the flood channel by taking for urban uses a large part of the floodplains. This floodplain width reduction (encroachment or contraction) implies a perturbation of the equilibrium (more specifically, a degradation), as demonstrated analytically and experimentally by Vanoni (1975), yet this is only one of the several causes of the degradation of a river bed (Galay, 1983).

As regards the delta, the relative importance of fluvial building and wave and tidal reworking determines the delta morphology and evolution (Bridge, 2003). The relevant maritime factors are reduced to wave action in the case of the Mediterranean sea

(no substantial tides). This wave action and its related currents produce a certain longitudinal coastal sediment transport, as well as a transfer of sand towards the open sea. The dominance of the fluvial or the maritime factor varies in space and time for a given delta. However, the simple statement made herein is that the greater the river sediment supply the more the delta will protrude into the standing water body, to equality of the maritime factor, and vice versa. Literature on delta evolution is abundant (e.g. Orton and Reading, 1993, Syvitski and Saito, 2007) and on river evolution as well (e.g. Rinaldi and Simon,

1998, Martín-Vide et al, 2010), but the connection between the two is less well known in physical terms, in spite of statistical approaches (Ibáñez et al., 2019, Xing et al. 2014). It is difficult to find data to evaluate the disturbance of river sediment supply on delta evolution, except a few cases such as the Mississippi river (Allison et al, 2012, Viparelli et al, 2015). A connection of this type is attempted in this research.

The retreat of beaches (specially in deltas) is a big concern in the Mediterranean region. The evidences of a long-term beach retreat in the Llobregat delta are presented first. This is a new set of data for this particular river, not presented so far. Then, the paper concentrates on the fluvial component, with the main objective of finding out whether the river channelization and encroachment has produced some retreat of the delta. Our method is to examine one by one the causes of change of river sediment yield, specifically of bed load. In this way, our focus is on what controls the coarse sediment yield of the river into

the sea, nourishing the beaches with sand (part of the coarse load). To know what controls the yield into the sea implies, as a consequence, to figure out which measures are more sensible in order to keep providing enough sand to the beaches. The management of the basin will benefit from this knowledge.

   Two natural hazards, flooding and fluvial erosion, are being addressed by the Llobregat case. The river has flooded its lower

valley and delta for centuries (Codina, 1971). Channelization during the last decades has brought a temporary relief to flooding, partly due to bed degradation, i.e. erosion of the alluvial bed as flow concentrates. Despite temporary reduction of flooding risks, this situation is not welcomed, actually, because serious consequences on groundwater levels and on riparian vegetation are to be feared. Regarding the consideration of fluvial and coastal erosion as a natural hazard, it should be borne in mind that sandy beaches on the Llobregat delta shoreline represent a precious natural resource in danger since the wave action is not

compensated anymore by sandy sediment yield coming from the river.

   Llobregat River is 163 km-long and drains an area of 4925 km$^2$ of the Northeastern Iberian peninsula, with its headland in the Pyrenees mountain range (fig.1). Archeologists have found evidences of human presence in the delta since Roman times (Marquès, 1984). The present delta ($\approx 100$ km$^2$) results from the Holocene transgression (6000 years ago, Ibáñez et al, 2019),

yet we are more interested in the delta evolution in the last century (within the so-call Anthropocene). The Latin name of the river was Rubricatus, which means dyed in red, in allusion to the color of its waters, probably because of its large fine sediment load. Moreover, Llobregat is today a gravel-bed river upstream of its delta, with a high bed load transport capacity. The delta

can be classified as sandy mixed load (bed and suspension) with only one distributary, following Orton and Reading, (1993). More river features and flood history will be given opportunely.


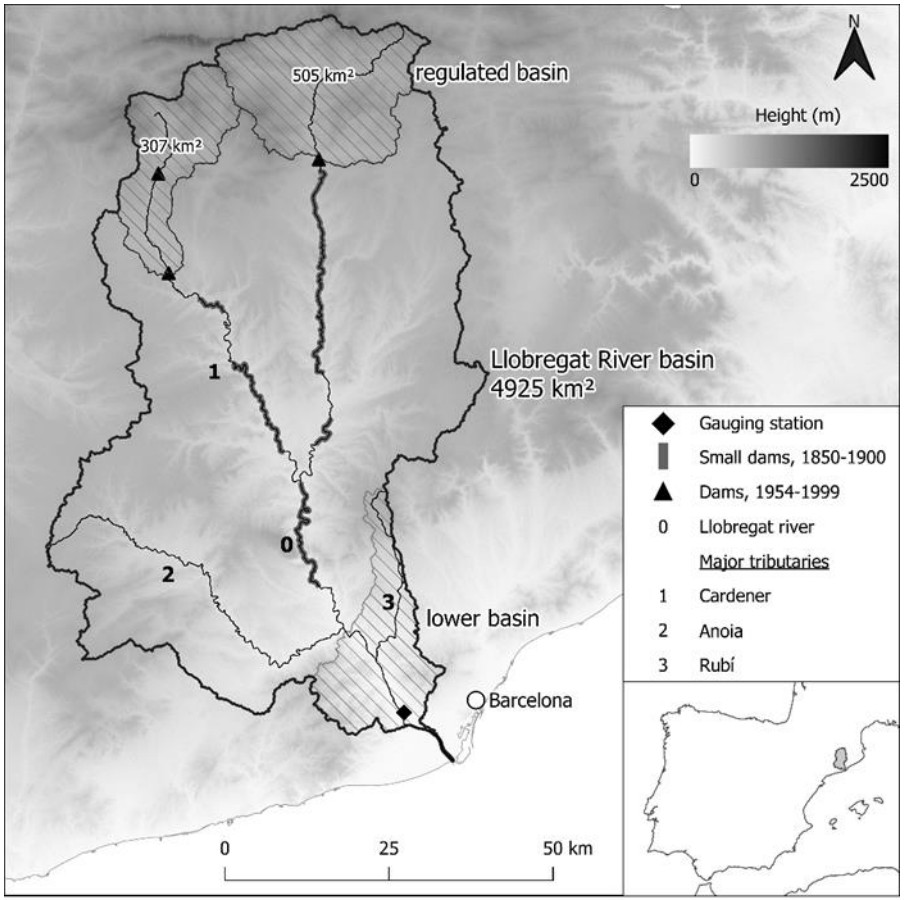

Fig.1. Location map. For lower basin see zoom in fig.4. For small dams see discussion section.

## 2 Beach retreat

Contrary to the delta building up in old times, it is heavily receding in the last century. Fig.2 shows the coastline in the area of the river mouth since 1891 until 1956, with point data for 1862 and 1907 (taking advantage of the mouth lighthouse, that was much inland at that time) and two intermediate lines in 1926 from a map and 1946 from an aerial photograph. Three more of them, dated in 1965, 1974 and 1981 show further receding of the coastline. The coast is a 24 km-long beach (fig.3), between a northern closed boundary (Barcelona harbor) and a partially open western boundary. The reach is a sedimentary unit

throughout the whole period 1891-1981. More recent photographs, such as the 2000 shoreline in fig.2, show the coastline much intervened by the enlargement of the northern harbor and the construction of dikes and of a second harbor at the western

boundary. In addition, dredging for beach nourishment has become normal in recent years. Due to these modern interventions, the present analysis is limited to the period 1891-1981 and more accurately to 1946-1981, although we will resort to other facts dated in the 19 century in the discussion. The current situation of the river mouth since 2004 is presented at the end of the

paper, not only for information but because it provides a kind of sediment closure

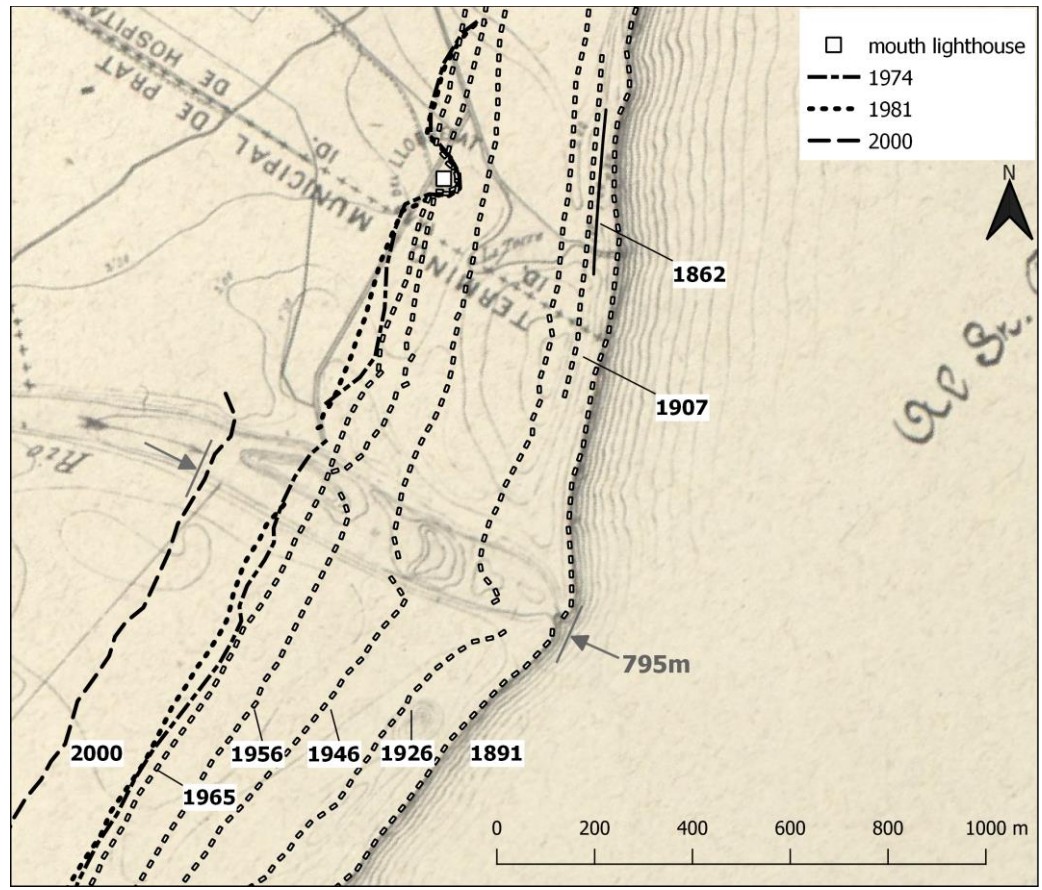

Fig.2. Coastline retreat in the Llobregat delta since 1891 (date of the map in the background, as well). Figure produced by authors using our own and freely available data from Institut Cartogràfic i Geològic de Catalunya (ICGC). The 1862 line

comes from Marcos (1995). The lighthouse was built as a watchtower in 1567. Drawings show its location well inland in the 17 century. It was turned into a lighthouse in 1852 to prevent ships to get stuck in the sandbanks of the river mouth.

The coastline change, either progradation into the sea or retrogradation inland (retreat), expressed in m, is summarized in fig.3 for the period 1946-1981 when photographs are good, almost complete in area coverage and the coast is not intervened yet.

The total change in these 35 years, discretized in reaches 1 km-long, is plotted against an abscissa x from West (left) to North (right), together with the change in the first and second decades (1946-1956-1965) to show temporal trends and oscillations.

An oval contour slightly protruding into the sea can be assigned to the length between x=15 and x=24 km, being the river mouth at x=21 km (see plan view in fig.2). In this 9 km-long reach, the delta has been receding in a coherent way, in the sense that the closer to the river mouth, the larger the receding, suggesting the key role of a decrease in the river sediment yield. This trend is quite common through different decades (fig.3). The beaches between x=0 and x=15 km, on the contrary, are mostly prograding, yet the temporal and spatial fluctuation in this area has been more noticeable. The delta is then composed of two sedimentary subunits. It is even conjecturable an old river mouth around km 10

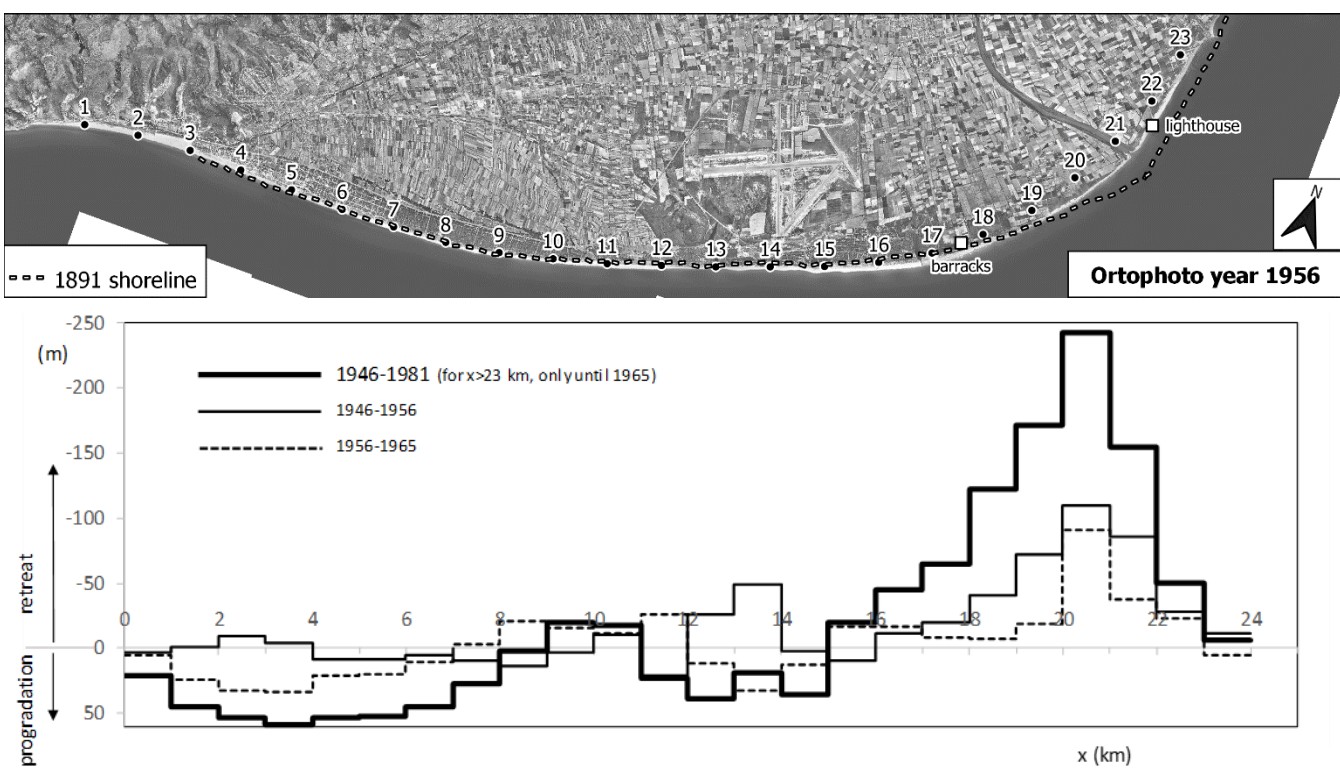

Fig.3. Above: the 24 km-long sedimentary unit (delta), produced by authors using freely available data from ICGC. Below: Total change in m perpendicular to the coastline in ordinates (progradation +, or retreat – ) along the coastline above in the period 1946-1981 and in two decades within it. Apart from the lighthouse in fig.2, the history of the barracks at x≈ 17.5 km in 1844 is also known: since that date until 1934 the coastline prograded 179 m (Paladella and Faura, 1935).

## 2.1 Volume calculation

The sand grainsize in the long delta beach is around 280 μm (Gracia and Calafat, 2019). The longitudinal sediment transport goes from North to West, with a transport capacity in the range 10.000—75.000 $m^3$/yr (CIIRC, 2010). The closure depth of the beach platform in the delta, i.e. the depth under sea level involved in the sediment transport shaping the beaches, is around 6.35 m. In turn, the berm height above sea level, involved as well, goes from 0.9 to 1.4 m (CIIRC, 2012). Then, every km of

beach in the coastline, either prograding or retreating 1 m, means a sand volume of $\approx 3.500$ m$^3$, respectively deposited or eroded (Digital Shoreline An. Sys. by U.S.G.S., Himmelstoss et al, 2018). The computation of sand volumes, by multiplying the change in m (fig.3) by 3.500 m$^3$/km, produce gross volumes, converted into net volumes, by deducting some 35% of voids. The calculation yields a deficit of 63.000 m$^3$/year in the north (x=15—24 km) and a surplus of 25.000 m$^3$/year in the beaches west of it (x=0—15 km). The temporal distribution of these net volumes over the four periods from 1946 until 1981 is (table

115 1):

| net volume ($10^3 \times$m$^3$/year) | 1946-56 | 1956-65 | 1965-74 | 1974-81 | 1946-81 |
|---|---|---|---|---|---|
| surplus, x=0-15 km | *—5 | +32 | (1)+21 | (1)+32 | **+25** |
| deficit, x=15-24 km | —84 | —54 | (2)—52 | (2)—30 | **—63** |
| balance (surplus vs. deficit) | —89 | —22 | —31 | +2 | —38 |

Table 1. Volumes of change of sand ($\times 10^3$ m$^3$ per year), distributed by decades and by region (oval delta in the north, x=15-24 km, and beaches west of it, x=0-15 km). * it is a deficit, actually, not a surplus, note the minus sign, (1) extended over 10
km instead of 15 km, (2) extended over 7 km instead of 9 km.

The deficit is larger than the surplus three times out of four in table 1. The negative balance (loss of sand) can be explained by the partially open western boundary (at x=0). The coastal longitudinal transport capacity cited above (net volume of 10—75$\times 10^3$ m$^3$/yr) seems capable, by order of magnitude, to take these amounts of sand from North to West beaches and even to
push part of it across the western boundary.

**3 River sediment yield**

One lacking piece in the balance of the coastal system is the sand sediment yield supplied by the Llobregat River, to which the core of this paper is devoted. Our objective is to ascertain to which extent the river sediment yield is important to the delta evolution, as the distribution of beach retreat in fig.3 suggests. Did the river yield decrease over the same period 1946-1981?
Do river yield figures compare with the volumes in table 1?, and which hydrological, hydraulic or sedimentary factors control the river yield? Similar to what has been done about the beach retreat, we will primarily use historical information on the river condition in 1946-1981, although discussion of the results will require to go back to the river condition in the 19 century.

The decrease of the sediment yield of a river to its delta may be due to different causes. Here we will consider: a) land use
changes including urbanization, b) the construction of large dams, so that reservoirs which: 1) trap sediment and 2) regulate flow, and c) river engineering works of any kind (mining included) on the channel and floodplains.

Cause a) affects primarily one component of the sediment load, the wash load, i.e. the fine sediment coming from anywhere in the basin. Cause b) affects all components of the sediment load but certainly its coarse fraction, which is more prone to get trapped in reservoirs than wash load. Cause c) in the Llobregat case since 1946 has been the progressive encroachment of the river by infrastructures (roads and railways) and its channelization against flooding with bank erosion control measures, in combination with some gravel and sand mining. These engineering works affect sediment load coming from the channel, composed of sand and gravel, not the wash load coming from the basin.

The causes of change of river sediment yield are analysed one by one in the next paragraphs, with special attention to bed load at the end.

**4 Land uses and urbanization (cause a)**

Land use changes in the Llobregat basin have been analyzed comparing the best past aerial photographs (1956) with a modern land use map (2009) (CREAF research center). The results are summarized in table 2, with aggregation of land uses in only three main categories: agriculture, forest and urban. The percentages for the whole Llobregat basin show a modest change consisting of a loss of agriculture land for the equitable benefit of towns (urban), on one side, and forest, which grow on the abandoned fields, on the other.

| | basin 4925 km$^2$ | | lower basin 343 km$^2$ | | tributary 3, 124 km$^2$ | |
|---|---|---|---|---|---|---|
| | 1956 | 2009 | 1956 | 2009 | 1956 | 2009 |
| agriculture | 35% | 22% | 43% | 8% | 45% | 9% |
| urban | 2% | 8% | 6% | 37% | 8% | 43% |
| forest | 63% | 70% | 51% | 55% | 47% | 48% |

Table 2. Land use change in the whole, lower basin and tributary 3 sub-basin in 1956-2009 (Prats-Puntí, 2018).

For the lower Llobregat basin, amounting 7% of the total basin area (fig.1), the loss of agricultural fields is more important and benefits more the urban area than the forest (fig.4). The lower Llobregat channel close to Barcelona is the most intervened reach. The case of the most urbanized sub-basin, the tributary 3 catchment (figs.1 and 4, table 2), shows a more marked reversal of shares between agricultural fields and urban areas. There is some channelization in this tributary but not any dam. Therefore, cause c) must have been dominant in the large bed incision reported in it since 1962 (Martín-Vide and Andreatta, 2009).

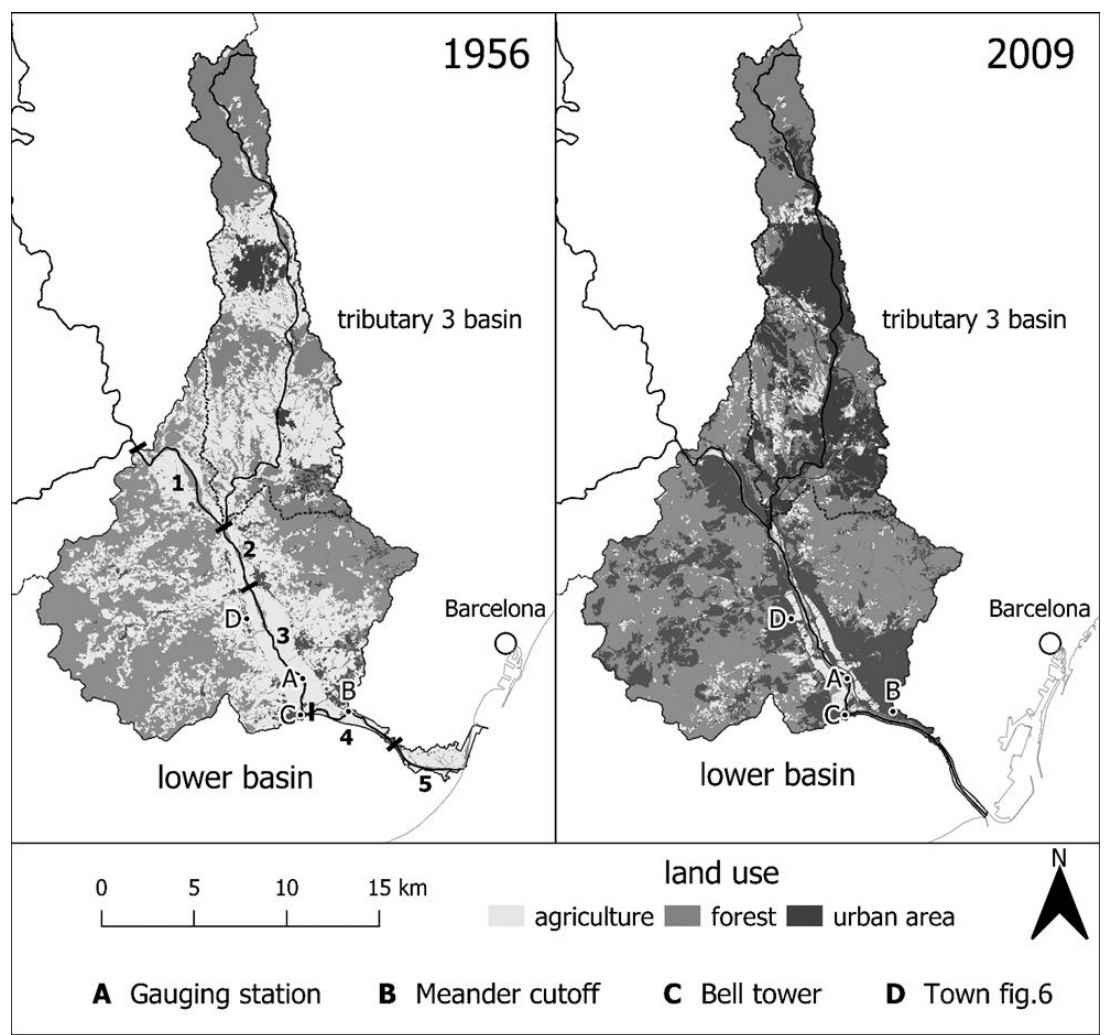

Fig.4. Land use changes in the Llobregat lower basin. Figure produced by authors using our own and freely available data from CREAF public research center. Note the added information on the lower river (reaches nr. 1 to 5 used in the analysis and reference points A to D mentioned in the text).

## 5 Reservoirs: sediment trapping (cause b1)

In the upper basin there are two areas regulated by dams (fig.1). One dam built in 1975 controls a 505 km$^2$ catchment, with a reservoir of 109 hm$^3$. The second dam, built in 1954 in tributary 1, controls a 307 km$^2$ catchment, with a small reservoir of 24 hm$^3$ (inside this second catchment, another dam with a volume of 80 hm$^3$ was built in 1999). Therefore, the area under hydrological control by large dams (the regulated basin) amounts to 812 km$^2$ since 1975, while it was of 307 km$^2$ in the years 1954-1975, that is to say a 16.5% and 6.2% respectively of the whole Llobregat basin. None of the three dams has any sediment by-pass device, nor are their bottom outlets able to pass or flush sediment, so far.

Sediment load coming from the regulated basin as wash load will be mostly trapped in the reservoirs, but the wash load component of the sediment yield, having dominant grainsizes in the clay-silt range (up to 62 μm), is not relevant for the coastline evolution, made of fine sand (280 μm). Regarding the load coming from the channels, ultimately trapped in the reservoirs, the drainage network density is similar all over the whole basin, but main rivers and tributaries are steeper in the mountainous regulated basins. Thus, the supply of coarse sediment from channels to the reservoirs is probably larger than the proportion of controlled catchment (16.5%), since most of the natural erosion comes from high elevation areas (Wilkinson and McElroy, 2007). Sediment supply is resumed in §9.

The previous reasoning must be extended to the flux of coarse sediment along the channel down to the river mouth. Dams produce a cut of the coarse sediment supply to the channel downstream, due to their sediment trapping capacity. This deficit travels downstream as a disturbance of incision (Martín-Vide et al, 2010), because supply is cut or reduced while transport capacity remains the same (this argument will be resumed in §9). Liébault et al. (2005) found a propagation velocity of 300-500 m per year for this disturbance (produced by reforestation in their case). In the south of the Iberian peninsula, Liquete et al (2005) showed that, although damming was active since 1970, up to a regulation of 42% of the basin areas, its effect was barely noticeable on the mouths of rivers with lengths 5—150 km by 2005. As the distance from dams to river mouth is more than 120 km in our case, it is highly unlikely that this disturbance has reached the lower Llobregat yet. In other words, the trapping of coarse sediment in the reservoirs since 1954 and 1975 must not have been relevant for the delta retreat yet, and neither for the period 1946-1981 of coastal retreat data (§2). In the long term, this trapping will come into picture. It is believed that this argument holds better for gravel than for sand.

## 6 Reservoirs: flow regulation (cause b2)

Reservoirs produce a second effect on sediment yield, through flow regulation, more precisely through peak flow attenuation. Once a reservoir stores water, the flow duration curve undergoes a reduction in peak flow along with an increase in low flow. These changes affect the sediment load coming from channels by means of two features of sediment transport: 1) the existence of a threshold for the initiation of transport, so that a reduction in peaks implies fewer days of flow above the threshold and so, more days with no transport, and 2) the non-linearity of bed load equations, in the sense that a certain reduction in flow means a higher reduction in bed load (f.e. 1/2 in flow but 1/4 in bedload, if bedload is proportional to the square of flow).

This effect can be assessed by comparing the flow duration curves with and without reservoirs. The period 2002-2018, after the last dam was built in 1999, is long enough to represent flows and reservoir management for computation of an average flow duration curve with reservoirs. Indeed, it is long enough to handle normal flows and annual floods, but not to take into account large floods, those occurring at return periods larger than, let's say, 10 years. Since no such large flood occurred in the period 2002-2018, the selected data describe normal flows and annual floods. The flow duration curve is done with the hourly data at the downstream-most gauging station (see fig.1 and 4). Moreover, this curve together with the contemporary measured daily levels at the reservoirs allow to compute a new flow duration curve without reservoirs, a "would-be" curve.

This is done by adding or subtracting the reservoirs volume variation in one day to the flow gauged at the station. The travel times of water from reservoirs to the station (22 h through main river and 20 h through tributary 1, according to the hydrographs of a real flood, fig.1) are the time lags between the volume variation at reservoirs and the discharge to be modified by addition or subtraction at the station.

Then, the comparison of flow duration curves with and without reservoirs assumes that the difference between the two are not much impacted by other hydrological and water resources changes, such as: *a*) water abstractions for irrigation and supply along the river, *b*) basin runoff due to land uses, and *c*) rainfall regime under the climate. It is not meant at all that flows are not impacted by *a*), *b*) and *c*), but only that their difference with and without reservoirs are not impacted. In other words, the reservoirs would have produced a similar difference in flow duration curves no matter the rainfall (climate), the runoff (land

use) and the abstraction (water use) had been. Under this assumption, the curve without reservoirs represents the state prior to 1954. Note that in this way we have circumvented the lack of any substantial river flow data prior to 1954. Moreover, if these data had existed, their use in comparison with the period 2002-2018 would have brought serious doubts of data homogeneity, just because abstractions, runoff and probably also rainfall regime have changed.

    The main result of this computation is that flow is higher without reservoirs than with them throughout the first 130 days; the

opposite happens over the rest of the year (fig.5). The representative discharge of the first day in the flow duration curve at the gauging station goes up from 259 m$^3$/s to 308 m$^3$/s and a similar, quite constant increase of $\approx 20\%$ extends to the first 100 days. The consequences of these results on sediment carrying capacity are discussed in §9.

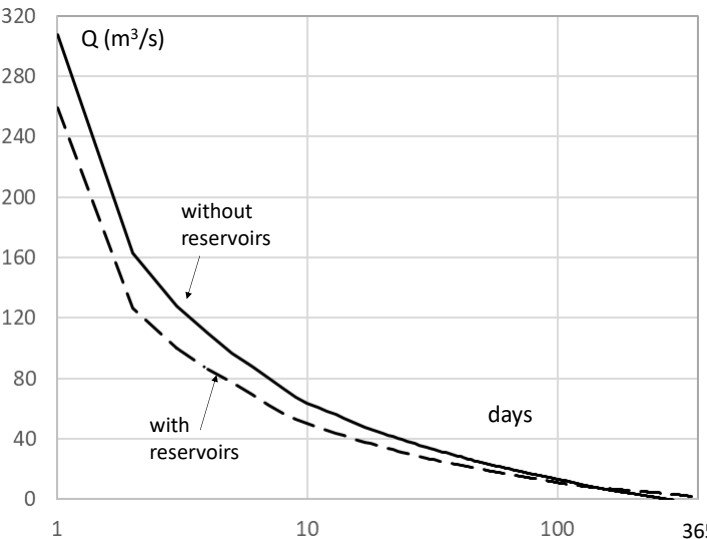

Fig.5. Flow duration curves at the gauging station (see fig.1 for location) with and without reservoirs. Log scale for days.

## 7 Data on the lower alluvial channel (information for cause c)

The lower Llobregat river stretches along 30 km from the junction of the last tributary (nr. 2, figs.1 and 4) to the delta mouth into the sea, with the gauging station located half way. It is the most intervened section of the Llobregat channel in the 20 century, luckily with the best archival records. Channel morphology (plan and long profile), grainsizes of the alluvium and the history of floods and engineering works (roads, railways, and flood defences) are obtained from these archives. Large floods in the lower Llobregat occurred in 1942, 1943 and 1944 ($\approx$1750 m³/s), 1962 ($\approx$2100 m³/s), 1971 ($\approx$3100 m³/s, the highest peak discharge) and 1982 ($\approx$1600 m³/s), within the period of analysis, broadly speaking (Codina, 1971). The 3-year period ending in 1944 is described in the documents as causing general aggradation. Just for reference, 1278 m³/s has been estimated as the 10-year return period flood and 3050 m³/s as the 100-year flood (Martín-Vide, 2007). Other floods in the 20 century occurred in 1907 ($\approx$2900 m³/s), 1919 ($\approx$1500 m³/s) (Codina, 1971) and 2000 ($\approx$1500 m³/s). The large floods in the 19 century will be mentioned in the discussion section.

For the sake of analysis, the 30 km-long channel is divided here in five reaches, 1 to 5, from up- to downstream (fig.4). In the first three (1-3), the channel used to be wandering within its wide valley floor, with incipient braids. In the last two (4-5), the river is rather a single thread, meandering, more stable channel running through the delta plains. Archival documents of different dates confirm this description. The corresponding bed slopes and mean grainsizes obtained from documents are gathered in table 3.

| reach | 1, valley | 2, valley | 3, valley | 4, delta | 5, delta |
|---|---|---|---|---|---|
| length (km) | 8.5 | 3 | 8 | 6.5 | 4 |
| slope 1946 ($\times 10^{-3}$) | 1.8 | 1.7[(1)] | 1.7 | 1.0 | 0.3 |
| slope 1982 ($\times 10^{-3}$) | 1.8 | 1.8 | 1.6 | 0.9 | 0.15[(2)] |
| $D_m$ (mm) | 21 | 15 | 17 | 8 | 0.7[(3)] |

Table 3. Slope and mean grainsize of the alluvial material for the five reaches of lower Llobregat. [(1)] is dated 1974 and [(2)] is dated 2016, actually, [(3)] additionally $D_{50}$=0.6 mm. Note the small slope change along time (1946-1982). (Prats-Puntí, 2018).

After table 3, the lower Llobregat is a 15-20 mm gravel-bed channel with a slope a little less than 2 per mil in the valley, which turns into a sand-bed river (much finer) with a much milder slope in the delta. This abrupt transition from a gravel-bed to a sand-bed stream typically goes with a sudden change in bed slope and stream morphology (Parker and Cui, 1998) such as wandering to meandering, as happens in our case. The important consequence is that the reach issuing sediment to the coastline is reach 5 with bed grainsize $D_{50}$ = 600 μm (table 3), similar to the grainsize on the beaches.

A cross-section representative of each date and each reach was drawn with the aid of aerial photographs and archival documents. One example is fig. 6 for reach 1 (see also sketch in fig.8, later, for a section in the border between reach 2 and 3).

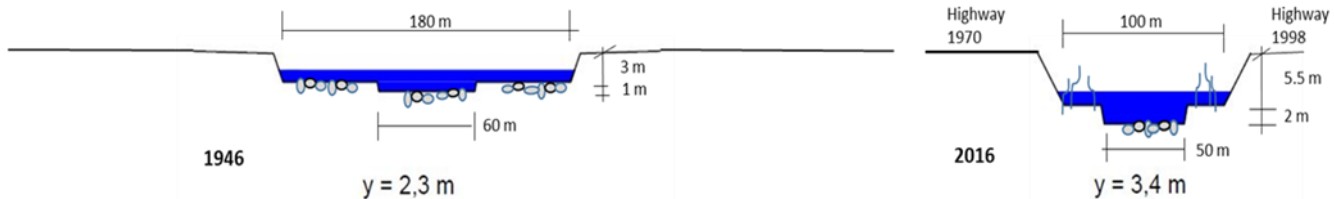

Fig.6. Cross-sections of reach 1 for the two extreme dates, 1946 (left) and 2016 (right). Alluvial widths are 175 m and 33 m respectively (table 4, later). Depth y drawn in the cross-sections corresponds to a discharge of 600 m³/s, within the first day of the flow duration curve.

## 8 Alluvial channel width and history of infrastructures (more information for cause c)

Width changes during (1946-1982) have been extremely large. Table 4 collects the alluvial bed surfaces in hectares obtained from the series of aerial photographs (§2). These surfaces are strictly alluvial, excluding the areas of early colonizing plants growing there. The average width shown in the table is the alluvial area divided by the reach length. Note the reduction to roughly half of the alluvial area in the period 1946-1981 (up to one third in reach 3). The current situation (2016) shows the last stage of the dramatic loss of alluvium, so far.

| reach | 1, valley | 2, valley | 3, valley | 4, delta | 5, delta | lower Ll. |
|---|---|---|---|---|---|---|
| length (km) | 8.5 | 3 | 8 | 6.5 | 4 | 30 |
| alluvial surface (Ha) / *average width* (m) | | | | | | |
| 1946 | 148 / *175* | 54 / *180* | 119 / *150* | 57 / *90* | 35 / *90* | 413 / *138* |
| 1956 | 86 / *100* | 33 / *110* | 57 / *70* | 42 / *65* | 25 / *62* | 243 / *81* |
| 1965 | 106 / *125* | 47 / *157* | 67 / *84* | 41 / *63* | 28 / *70* | 289 / *96* |
| 1974 | - | 49 / *163* | 53 / *66* | 43 / *66* | 30 / *75* | 175 / *81*[†] |
| 1981 | - | 30 / *100* | 41 / *51* | 54 / *83* | 30 / *75* | 155 / *72*[†] |
| 2016 | 28 / *33* | 18 / *60* | 29 / *36* | 23 / *35* | 77 / *190* * | 98 / *38*[††] |

Table 4. Alluvial surfaces and average widths of the strictly speaking alluvium in the aerial photographs. * this figure have to do with the new mouth (§14), [†]these figures extended to and averaged over the lowermost 21.5 km (reaches 2-5), [††]idem in the uppermost 26 km (reaches 1-4). For an example see fig.6 (Prats-Puntí, 2018).

This change is conspicuous for any observer of the river. For example, the river landscape prior to 1920 is compared to the present state in fig.7, both photographs taken from the bell tower of town C (see fig.4 for location). The same conclusion of a dramatic change is drawn from archival plans and documents. The widest, wandering Llobregat of 1946 seems to be related to the aggradation brought by the 1942-1944 floods.

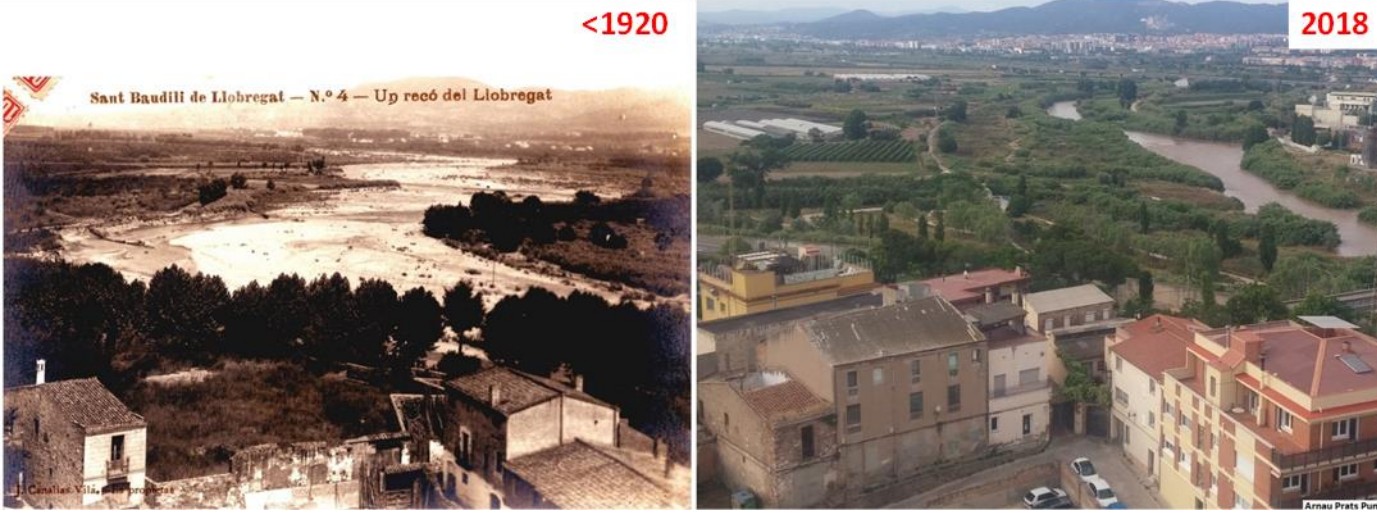

Fig.7. Pictures of the Llobregat looking upstream prior to 1920 (left, anonymous in Catalan National Archives) and in 2018 (right, by A.Prats-Puntí) from the same viewpoint on top of the bell tower in town C (fig.4).

These changes have been forced by the infrastructures serving the urban area of Barcelona. Reaches 1-3 make the main corridor
of roads and railways across the mountain range towards the plains where the city stands. Dates of opening of the main infrastructures are: 1970 for a highway (built as a dike) through the middle of the left floodplain; 1979 for a meander cutoff (fig.4); 1998 for the companion highway (another dike) through the middle of the right floodplain, followed by the railway attached to the riverine side of this dike (and 2004 for the new river mouth into the sea). Fig.8 is a close view of a particular section around town D (fig.4 for location). It shows that the highways are also flooding dikes (or levees), which encroach upon
the floodplains.

This calendar of works suggests that only the last four rows in table 4, showing a reduction of average alluvial width from 96 m (1965) to 72 m (1981) and ultimately to only 38 m (2016) are attributable to the main infrastructures, which have cut off roughly half of the floodplain width at least.

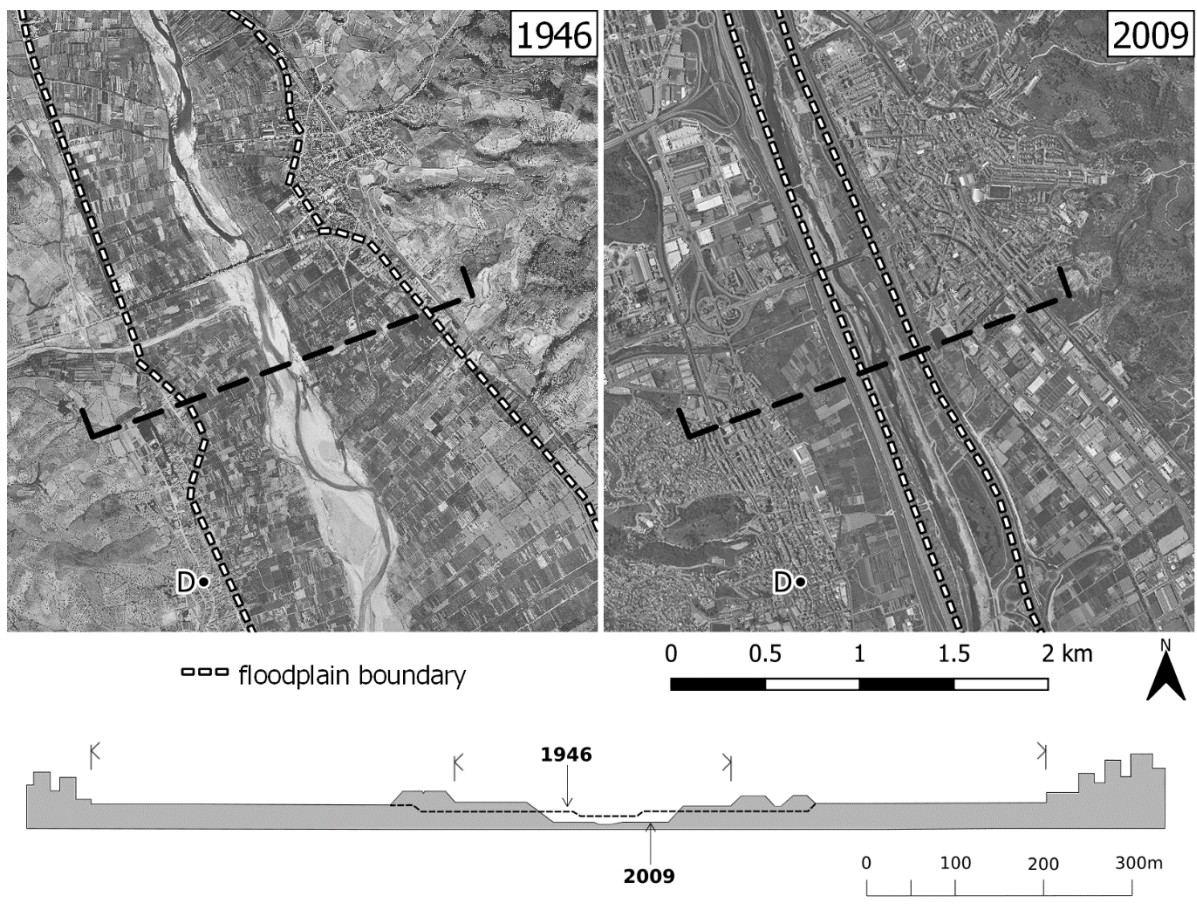

Fig.8. Above: plan view in 1946 (left) and 2009 (right) of the river around town D (see fig.4 for location) in the overlapping of reaches 2 and 3, produced by authors using freely available data from ICGC. The bridge in the middle of the left picture (see fig.10, later) failed in 1971. Below: crosssection in reach 3 taken through the dashed line above (see fig.6 for more cross-sections)

Some other works are worth mentioning. After the 1944 and 1962 floods, several river training works of lesser scope were executed. Gravel mining operations in the active channel were still minor in 1956, larger in 1965 and their heyday was in 1974, while they were declining in 1981. Most of the mining pits were located in reaches 2 and 3. Unlike the 1970 left highway, the engineering works for the 1998 highway and railroad included the digging of the channel, from the left dike to the right dike, to allow for flow in case of floods.

**9 River engineering: supply sources (cause c)**

As presented in §5 for the sediment trapping by dams, the bed material transport of a river reach is the balance between the supply from upstream and the carrying capacity of the reach (Einstein, 1964). Focusing now on supply, table 4 provides metrics to the bed material source of supply. High flows and floods are able to pick particles from those alluvial sources, which in this way keep being alluvial, as seen in the aerial photographs. Thus, table 4 is useful as indicator of the change of supply in time within the lower Llobregat. For example, the alluvial bed surface in reach 1 goes down from 148 to 86 Ha in the decade 1946-1956 (or from 175 to 100 m in terms of average alluvial width), so that the likely supply to reach 2 from reach 1 is probably reduced in the same proportion. The supply to reach 1, the first in the row, from further upstream is treated in the discussion.

Unlike the effect of the upland dams, the disturbance of this supply cut is likely able to affect the lower Llobregat, at least the next reach downstream of the one considered, if a disturbance velocity of hundreds of m per year (for ex. 500 m/yr, Liébault et al, 2005) is reasonable. In one decade then, such as 1946-1956, reach 2 would be affected by the supply cut in reach 1, and so on for the next reaches and decades. Unlike the case of dams again, this disturbance is not necessarily one of degradation, because each reach downstream suffers a comparable reduction of alluvial bed as the reach upstream. For example, reach 2 goes down from 180 to 110 m in width (table 4) in 1946-1956, at the same time as reach 1 reduces its own width from 175 to 100 m. The reduced supply due to a narrower alluvium upstream finds a narrower cross section downstream to carry it further downstream. Whether the difference of supply and carrying capacity is positive or negative, the result will be aggradation or degradation in reach 2 (this argument will be resumed in §9).

Carrying capacities are dealt with next, but the point to be retained now is that the changes of alluvial area in the lower Llobregat are able to control the sediment yield of the river in a period of three to four decades (1946-1981) and even more in the lapse of time until present (1946-2018).

**10 River engineering: carrying capacity (cause c)**

Assuming uniform flow and bed shear stress proportional to hydraulics radius and bed slope (table 3), we have applied the Meyer-Peter and Müller (MPM) equation (Wong and Parker, 2006) to the cross-sections of each date and reach, for each hour of the flow duration curve, with and without reservoirs, in order to get unit solid discharges, which multiplied by the alluvial widths produce table 5

The ratio of carrying capacity with and without reservoirs is 0.62 for reaches 1- 4 and 0.73 for reach 5, on average (table 5). In other words, flow regulation by reservoirs is responsible for a reduction of carrying capacity amounting to 38% in most of the lower Llobregat today (reaches 1-4), which is quite more that the reduction of discharge in the flow duration curve of the present flow regime with reservoirs ($\approx$ 20 %, §6).

| reach | 1, 8.5 km | 2, 3 km | 3, 8 km | 4, 6.5 km | 5, 4 km |
|---|---|---|---|---|---|
| 1946 | 5.6 | 12.9 | 9.6 | 12.0 | 12.7 |
| 1956 | 7.5 | 11.6 | 8.9 | 14.2 | 16.1 |
| 1965[†] | 7.5 | 16.2 | 9.8 | 14.1 | 16.3 |
| 1974[††] | - | 7.5 / 4.6 | 3.8 / 2.3 | 7.9 / 4.9 | 13.7 / 10.7 |
| 1981 | - | 8.6 / 5.3 | 3.0 / 1.8 | 6.2 / 3.7 | 13.5 / 10.5 |
| 2016 | 6.3 / 3.9 | 15.6 / 9.8 | 8.7 / 5.4 | 11.4 / 7.9 | 1.5*/ 0.95* |

Table 5. Carrying capacity ($\times 10^3$ m$^3$/yr) of the five reaches and all years. The underlined figures are the capacity with reservoirs. [†]computed with none of the reservoirs in operation, [††]computed with the three reservoirs in operation, *these figures have to do with the new mouth (§14) (Prats-Puntí, 2018).

Carrying capacity computed in this way is proportional to the alluvial width, but also it is affected by the depth increase, because this implies an increase in shear stress (1.80 times higher in 2016 than in 1946 in the case of fig.6).

**11 Estimation of the real coarse sediment transport**

The balance between supply and carrying capacity states that if the former is larger than the latter, aggradation occurs and the amount conveyed further downstream equals the carrying capacity only, not the supply. If the opposite happens, the amount conveyed is the supply plus material from the bed (and so, degradation occurs), as long as the alluvium is not exhausted but available, tending to the carrying capacity at the most. If the two quantities are equal, equilibrium holds.

The delta was steadily retreating before 1946 (fig.2), the date of the first quantitative, extensive information (photographs), so that supply and carrying capacity changes from 1946 can be seen as disturbances to this state. An attempt to this "disturbance" analysis is table 6, which combines data of table 4 for supply and table 5 for capacity, in the form of percentages with respect to their 1946 values, either alluvial area (Ha), the surrogate of supply, or computed carrying capacity (m$^3$/yr). After table 6, in 1956-1965 capacity exceeded supply because it went higher than in 1946 or at least kept quite high, but supply dropped significantly (so, degradation was likely), whereas in 1974-1981 supply exceeded capacity because it was capacity that dropped very much while supply kept still at the previous level (so, aggradation was likely). The degradation observed in 1956 for the whole lower river probably followed the 1944 flood aggradation (§7).

| coarse sediment supply > or < bed material carrying capacity (% to 1946) | | | | | |
|---|---|---|---|---|---|
| reach | 1 | 2 | 3 | 4 | 5 |
| 1946 | 100 = 100 | 100 = 100 | 100 = 100 | 100 = 100 | 100 = 100 |
| 1956 | 58 < 135 | 61 < 90 | 48 < 93 | 74 < 118 | 71 < 127 |
| 1965 | 72 < 135 | 87 < 126 | 56 < 103 | 72 < 118 | 80 < 129 |
| 1974 | - | 91 > 36 | 45 > 24 | 75 > 41 | 86 > 84 |
| 1981 | - | 55 > 41 | 34 > 19 | 95 > 31 | 86 > 83 |
| 2016 | 19 < 70 | 33 < 76 | 24 < 56 | 40 < 66 | 220 > 7 |

Table 6. Comparison of the amounts of coarse sediment supply (figure left) and bed material carrying capacity (figure right) by reaches and years, with reference to a level 100 of both quantities in 1946. The underlined values in table 4 (with reservoirs) are used; symbols †, †† and * in table 5 apply here too. Dark grey boxes mean likely aggradation (>), light grey likely degradation (<).

## 11.1 Routing algorithm

As stated above, the volume dispatched downstream is the capacity if supply > capacity and the capacity at most if supply < capacity. Then, the logical operation < or > in any row of table 6 would allow to transfer amounts in m$^3$/yr (capacities) to the next period and reach, serving there as supply (case >) or supply at most (case <), to be compared to capacities in a consistent way (same unity, m$^3$/yr). This lapse (one decade) and step (one reach) have been justified in §5 and §8 with the velocity of the disturbance created by a cut of supply. This kind of algorithm is applied to produce table 7 by starting with the 1956 row in table 6 and using the data of table 5. Regarding the "boundary" data, i.e. year 1946 and reach 1, we assume for the moment that carrying capacities are dispatched quantities to the next reach and period.

Table 7 provides an estimate of the sand sediment yield into the sea in the last column, i.e. $\approx 16 \times 10^3$ m$^3$/yr in the period 1956-1965 but $\approx 10 \times 10^3$ m$^3$/yr in 1974-1981. If the river had not been regulated by dams, the yield in 1974-81 would have raised to $\approx 13.5 \times 10^3$ m$^3$/yr (see table 5), i.e. $3.5 \times 10^3$ m$^3$/yr more without dams than with them.

| reach | 1 | 2 | 3 | 4 | 5 | to coast |
|---|---|---|---|---|---|---|
| 1946 | 5.6 | 12.9 | 9.6 | 12.0 | 12.7 | → 12.7 |
| 1956 | 7.5 | 5.6 < 11.6 | 12.9 > 8.9 | 9.6 < 14.2 | 12.0 < 16.1 | → <16.1 |
| 1965† | 7.5 | 7.5 < 16.2 | 11.6 > 9.8 | 8.9 < 14.1 | 14.2 < 16.3 | → <16.3 |
| 1974†† | - | 7.5 > 4.6 | 16.2 > 2.3 | 9.8 > 4.9 | 14.1 > 10.7 | → 10.7 |
| 1981 | - | 5.3 | 4.6 > 1.8 | 2.3 < 3.7 | 4.9 < 10.5 | → <10.5 |
| 2016 | 3.9 | 9.8 | 5.3 < 5.4 | 1.8 < 7.9 | 3.7 > 0.95 | → 0.95 |

Table 7. Coarse sediment transport ($\times 10^3$ m³/yr). The quantities at the right-hand side of symbols > or < are capacities from table 5 (with reservoirs), those at the left-hand side are supplies transferred. Dark and light grey boxes the same meaning as above. Dotted lines with arrows mean transference to the next reach and decade and arrows mean transference to the coast. The symbols †, †† and * in table 4 apply here too.

By comparing table 7 with table 1, the computed annual river yield decreasing from $\approx 16\times10^3$ in 1956 to $\approx 10\times10^3$ m³/yr in 1981 is found to be a substantial factor for the delta evolution. It is of the same order of magnitude but lower than the delta balance in 1946-1981 ($-38\times10^3$ m³/yr). Its variation of $\approx -6\times10^3$ m³/yr between 1956 to 1981 due to the river encroachment by infrastructures is less substantial, but it still accounts for $\approx 16\%$ of the balance. The role of the regulation by dams, i.e. a variation of $\approx -3.5\times10^3$ m³/yr, accounts for some 9% of the balance. It must be recalled that the computation is based upon mean flows and annuals floods, not including large floods, whereas the delta evolution (§2) encompasses all phenomena. The role of large floods is explored next.

**12 Incision in large floods**

Just after the building of the left dike (highway) in 1970 (§8), the largest flood of the 20 century in 1971 caused a general bed degradation in reaches 2 and 3. An historical bridge close to town D in fig.8, failed due to that event (Batalla, 2003). Similarly, the 2000 flood with a peak of 1500 m³/s (§7) came just after the construction of the right dike in 1998. In addition to the dikes, the river channel had been dug to increase hydraulic capacity. The 1998 "as-built" bed profile has been compared with a survey after the 2000 flood, resulting in incisions of 0.6 m along 2.5 km of reach 1, a minor amount in reach 2 and 0.5 m along 3.0 km of reach 3. Therefore, the volume scoured was $\approx 70\times10^3$ m³ in 1 and $\approx 55\times10^3$ m³ in 3. A sum of $\approx 125\times10^3$ m³ was issued by the valley reaches (1-3) to the delta (4-5) and ultimately to the coast. It is clear, therefore, after comparing this amount to those in tables 1 and 7, that large floods may be dominant in the sediment yield. The river bed supplied particles, as theory claims, at the cost of incision. By inductive reasoning, incision will happen again as long as the alluvium does not get exhausted.

Our computation of $16\times10^3$—$10\times10^3$ m$^3$/yr is a large underestimation in years with large floods, which can act as pulses driving the delta evolution. It is a challenge to know to what extent this is so, with our scarce data. One can hardly make a count of 9 large floods similar to the 2000 one in the 20 century (§7), so that a rough average amount 'per year' of the century would be $11\times10^3$ m$^3$/yr. Thus, the total yield in the long term, including large floods, would double our computation for normal flows and annual floods. In this way, the river yield ($+32$—$20\times10^3$ m$^3$/yr) would be close to match, with opposite sign, the delta balance ($-38\times10^3$ m$^3$/yr), both in the range $20$—$40\times10^3$ m$^3$ in absolute value.

## 13 The role of channelization on delta retreat

Table 8 shows the high variability when considering the same comparison by periods of analysis (decades). Deficit at delta (from table 1), computed river change (table 7) and flood occurrence (§7) are compared. Most interesting is the maximum delta retreat in a decade of no floods (1946-1956), after years 1942-1943-1944 of large floods and general aggradation (even causing an increase in river sediment yield $<+3,4\times10^3$ m$^3$/yr afterwards). This suggests that the "normal" river yield is largely insufficient to counteract wave action. On the contrary, the agreement in the last period (1974-81) suggests that floods larger than, but similar to, the 10 year return period (§7) can be sufficient to keep the delta almost at balance. However, the middle decades (1956-65 and 1965-74) contradict the last suggestion, since larger floods, close to the 50 year return period, did not produce delta aggradation, not even a balance, but a retreat. This discrepancy highlights the obvious role of the storms as special events of wave action on the sea side, similar to floods on the river side.

Nevertheless, we can extract more information from table 8: the difference in the balance between decades with floods (the last three) and without floods (the first), which amounts to the range $58$—$91\times10^3$ m$^3$/yr for the three last decades, confirms the order of magnitude of the river yield by one large flood ($\approx 100\times10^3$ m$^3$/yr), already obtained with data of the 2000 flood. More importantly, note that the largest impact of the encroachment by infrastructures (channelization) in the period 1965-74, depriving the delta of $5.6\times10^3$ m$^3$/yr (at most), produces an increase of the delta deficit in a similar amount of $9\times10^3$ m$^3$/yr (difference 31-22), both with respect to the period 1956-65. Thus, the contribution of channelization to the delta retreat, that seemed absent or hidden, can be evaluated as a portion $< 5,6 / 31$, i.e. up to 18% of the total balance, similar to $\approx 16\%$ by using average figures for the whole period 1946-1981 (§11.1).

| | 1946-56 | 1956-65 | 1965-74 | 1974-81 |
|---|---|---|---|---|
| balance at delta ($10^3\times$m$^3$/yr) | —89 | —22 | [2]—31 | [2]+2 |
| computed river change ($10^3\times$m$^3$/yr) | $< +3.4$ | $\approx 0$ | $>-5.6$ | $\approx 0$ |
| any large flood? if so, Q (m$^3$/s) | no | 2100 | 3100 | 1600 |

Table 8. Comparison of table 1 and the differences in table 7, for the balance at the delta and the change in sediment yield. Flood discharges come from §7. [2] see table 1.

**14 The new mouth and closure of the computation with real data**

A new mouth of the Llobregat river, moving the channel southwards to let more room for the port of Barcelona, was opened in 2004 (see fig.4). It is a very wide canal (width from 105 m inland to 215 m at the end) with a flat bottom excavated at elevation —2 m (below sea level). The new width is more than twice the original one (table 4), so that its carrying capacity (table 5) and, then, its sediment transport (table 6) go down one order of magnitude . The bottom elevation is also much lower than the original one. Therefore, it was prone to alluviation and silting up.


It was not a surprise, then, that a survey in 2009 disclosed a sedimentation of $700 \times 10^3$ m$^3$ in the new mouth (or 0.5 m of aggradation throughout), i.e. an average of $140 \times 10^3$ m$^3$/yr in 2004-2009. Material trapped in the new mouth is not only sand, of course, but the finer suspended load as well. In other terms, it is the sum of bed load and suspended load, or the sum of bed material load and wash load, i.e. a total load. Moreover, the suspended load has been estimated through measurements of

concentration of suspended sediment in the above mentioned gauging station (fig.4 for location) in 1995-2002, resulting a total suspended yield of $\approx 90 \times 10^3$ m$^3$/yr  (Liquete et al, 2009, assuming a sediment density of 1.1 t/ m$^3$ for fresh sediments, Batalla, 2003). These daily measurements could not monitor in detail the 2000 flood. The comparison of these figures of suspended and total load proves that the bed load component is not negligible.

The ratio of the bed load computed above (for years with no floods, i.e. $10—16 \times 10^3$ m$^3$/yr) and the total sediment load trapped in the new mouth is $\approx 10$ %. For six Mediterranean rivers: Ebro (Spain), Rhône and Var (France) and Arno, Pescara and Po (Italy), this ratio goes from 2% to 17% with an average of 7%. For the subset of Arno, Pescara and Var, the most similar in size to Llobregat river, the average ratio is 9% (Syvitski and Saito, 2007). This result brings confidence to the computation of this paper on the grounds of: *i)* the total load trapped in the new mouth, and *ii)* the typical bed load to total load ratio in

Mediterranean rivers of similar size.

**15 Discussion**

The bed-material yield in the last decades has been influenced by the channelization works in the lower Llobregat, which is close enough to the sea for their disturbance to be felt in the delta. The source of alluvial bed sediment got reduced from its 1946 level to just 38% of it in 1981 and to just 22% in 2016, in reaches 1-3 (table 4). The channelization reduced also the

sediment carrying capacity (tables 5-6), for example to 67% in reaches 1-3 (2016) with respect to 1946. This carrying capacity determines the actual (computed) sediment yield, going from $16 \times 10^3$ m$^3$/yr in 1956 down to $10 \times 10^3$ m$^3$/yr in 1981. This amount means some 10% of the total sediment yield, measured accidentally in a dysfunctional new river mouth. The agreement with the literature on the subject of bed load to total load ratio confirms the computation. All this is based on normal flows and annual floods, whereas large floods exceed the previous amounts by large (one order of magnitude). An estimate of another

$10×10^3$ m³/yr in average over a century, with the discussion on their crucial role in driving the delta balance (either retreat or aggradation) in 1946-1981, is included in §13.

The customary assumption of a "steady river" (no floods) by coastal specialists is equally wrong as the customary assumption of a "steady sea" (no storms) by river specialists. It could even be argued that the equilibrium of a delta is elusive, since the
delta either progrades, in case of floods and no storms, or retreats, in case of storms and no floods. The fluvial input to the delta (sediment yield, notably in high flows and floods) is controlled by intrinsic river variables, such as alluvial width and bed gradient, that have nothing to do with intrinsic coastal variables, such as beach profile, that respond to the maritime input to the delta (wave energy, notably in storms).

The sediment yield to the delta has not been reduced more heavily so far, because alluvial beds have provided much material, instead of the alluvial plains excluded from the channelized river, at the cost of incision in several reaches. Since its opening in 2004 the new mouth is further hampering the exit of sand to the coast because it is acting as a sediment trap, in such a way that the current yield is indeed reduced in one order of magnitude to ≈$1×10^3$ m³/yr (table 7). The likely future exhaustion of the bed in the channelized river together with the sediment trap, worsened if it is dug out for maintenance, is a future scenario
of more severe sediment cut for the delta and its beaches.

It has been demonstrated that the sediment trapping at the dams may not be as influential on the coarse sediment yield as the effect of flow regulation due to them, which implies a reduction of carrying capacity amounting to 38%. However, some moderate effects of sediment trapping at dams should appear in the long term. A consequence for a management aimed at
providing sand to the beaches is that it is more effective a step back from the channelization than the efforts to pass sediment at dams. This statement points to what controls the coarse sediment yield in the river.

However, despite all the analysis shown so far, the influence of the modern river channelization on the delta evolution is overrun by a much larger long-term trend of the Llobregat delta, which seems irreversible as we will see. In fact, the
contribution of the channelization to the total retreat in the period of analysis, 1946-1981, has been evaluated over 16—18%. The retreating trend was clear in fig.2, updated several times to add new historical data while the effect of channelization was being analyzed. The most advanced delta coastline must have occurred around the turn of the 20 century, between 1891 and 1907. The question is why the delta was prograding in the 19 century, at least since 1862, but retreating continuously during the 20 century. Is there any explanation for the trend shift around 1900?

Case-studies of rivers in southeastern France (Liébault and Piegay, 2002) suggest that a reforestation policy in the last 150 years, applied to Catalan basins as in the French examples, may be influential in narrowing river channels and so, indirectly, in the retreat of deltas. However, the decrease of sediment sources (less agriculture and more forest) seems very modest in this

case (table 2), even more modest in the context of recent research that proves a weak signature of deforestation on delta size, because fine sediments contribute little to delta progradation (Ibáñez et al, 2019).

A second reason stems from a particular hydrological regime in the 19 century. Following documentary research, the period 1830-1870 was marked by a high frequency of floods in the Llobregat and other rivers of Mediterranean Catalonia (Llasat et al, 2005; Barriendos et al, 2019). The most severe floods occurred in 1837, 1842, 1853 and 1866 (Barriendos and Rodrigo, 2006). The 20 century has been less active: 6 catastrophic events in the 19 versus only 1 in the 20 (Llasat et al, 2005). A natural origin of this anomaly is accepted in the literature on the grounds of its temporary course and the corresponding climatic oscillations between several European regions. It can be assumed that these flood pulses produced an advance of the delta.

A third reason is the development of garment factories on the banks of the Llobregat river to profit from waterpower, in the 19 century (Alayo, 2017). This can be asserted for 91 factories in the middle reaches of the river (see "small dams" sign in fig.1), consisting of a diversion dam with average height of 4,2 m ± 2,9 m (standard deviation). Some 62% of them were built between 1850 and 1900 and most are still in operation as small hydro plants. More specifically, fig.9 is the graph of the cumulated height (m) versus the date of the insertion (calendar years) of small dams in the river. Following the progressive dam insertion, the span in height that keeps free for flow of water and sediment in the river profile is reduced accordingly. Recalling that the bed load carrying capacity is a monotonically increasing function of this free span, fig.9 also serves as a surrogate of the reduction in carrying capacity over the years. These 91 small dams date from 1816 till 1963 and stand from 4 to 100 km away from the upper border of the lower Llobregat reach (fig.1). The delayed effect of the farthest dams, and the quick effect of the closest, in the way to reach this border is taken into account by a disturbance velocity. The graphs for velocities 2 km/year, 1 km/year and 0.5 km/year are plotted in fig.9. Note that the latter has been used in the sediment routing through the five reaches of the lower Llobregat in §11.1 These graphs express the pace of the decrease in sediment supply at this border due to the space and time dispersion of the factories.

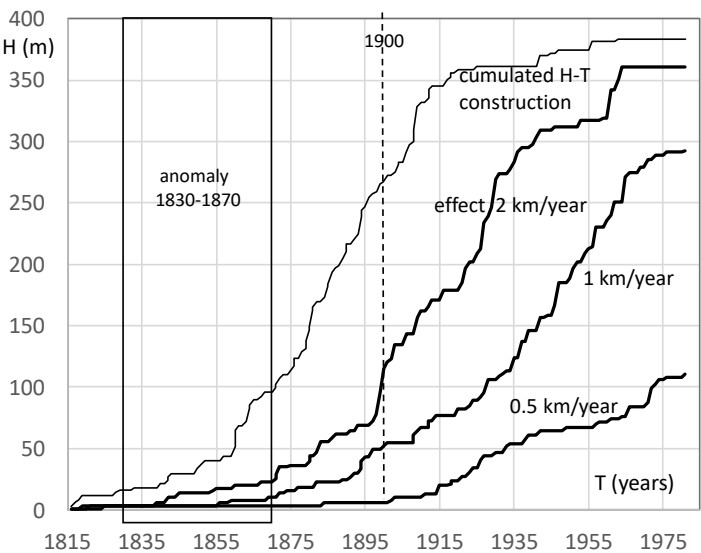

Fig.9. Cumulative height H (m) versus calendar date for the installation of factories in the middle reaches of the river (data in Alayo, 2017), and its effects at the upper border of the lower Llobregat, under three assumptions of disturbance velocity.


Two points are worth of discussion in fig.9: *i*) the hydrological anomaly of 1830-1870 finds the middle reaches of the river before the heyday of the garment factory building; therefore, the severe floods of this period must have brought large amounts of sediment to the lower Llobregat, and *ii*) the increasing effect of factory building on the sediment supply to reach 1 spreads throughout the 19 and 20 centuries, including the period 1946-1981 of our main analysis, and even beyond; the turn of the

century (1900) may be spotted as the fastest increasing supply cut in case of a 2 km/yr disturbance (or the incipient cut for a 0.5 km/yr disturbance) in order to explain the shift from progradation to retreat in the delta. Obviously, the recovery of free span in height in the middle river by removing small dams would be effective to increase the sediment delivery to the delta, in the long term (Ibisate et al., 2016).

In the event of a more active Llobregat in the middle years of the 19 century, and mostly free of factories in the middle reaches, the alluvial channel in the lower river should have been much wider at that time. Very fortunately, two plans of the lower Llobregat at reach 3, dated 1846 and 1854 just in the years of the hydrological anomaly, do exist in the National Archives to check our hypothesis. They can be scaled by means of landmarks in towns C and D and specially thanks to the historical bridge close to D that failed in 1971 (§12). Moreover, fig.10 is a photograph dated 1866-1867 of this bridge, a very telling picture of

the largest alluvial width known and the plenty of sand and gravel there at that time, completely lost today. The average widths within reach 3 from the two plans are 272 m (both 1846 and 1854), with maxima of 447 m (1846) and 579 m (1854) and minima of 155 m (1846) and 123 m (1854). Compare this with an average width of 150 m for reach 3 in 1946 (table 4). The heyday of the sediment yield to the delta was the middle of the 19 century. In 1900 things had started to change.

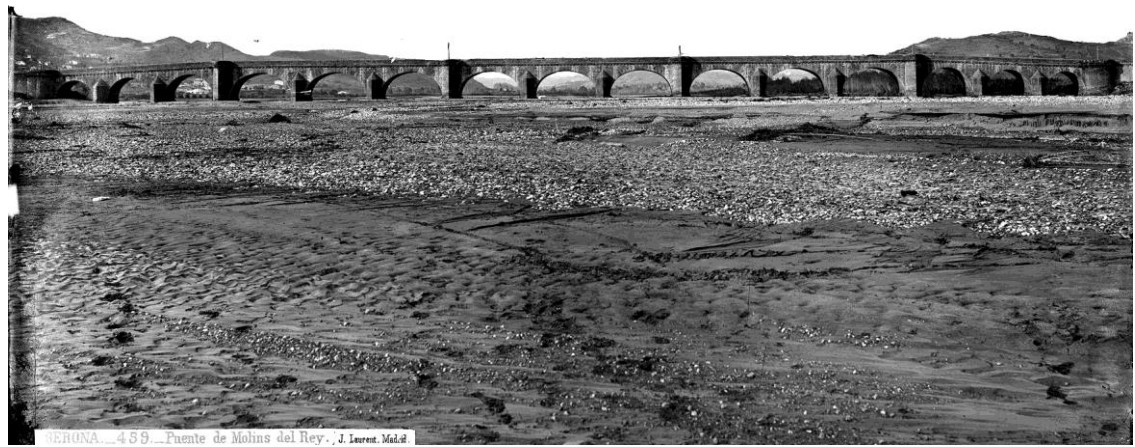

Fig.10 Bridge close to town D, shot by well-known French photographer Jean Laurent probably in 1866-1867. The only

bridge in lower Llobregat at that time had a total length 334,36 metres, with 15 arches, the central 9 of which spanning 19,22

m each. It failed in 1971. Note the extremely wide alluvial area full of sand and gravel.

**16 Conclusion**

The decrease in coarse sediment yield, causing the continuous retreat of the Llobregat delta throughout the 20 century, must

be attributed in some 80% to the frequency and intensity of large floods ("anomalous" hydrology) in the 19 century and the

large number of small dams built at that time, while the contribution of the land use change is minor. For the first reason

(hydrology), the retreat is probably irreversible. Modern encroachment by infrastructures (from 1970 to day) in combination

with flow regulation by large dams (1954 to day) explains the remaining 20%. The future is challenging in view of the new

mouth (2004 to day), the depletion of the bed alluvium (by floods under river encroachment rather than by mining), the

remaining effect of the past small dams and the long-term effect of modern large dams, let alone in view of the climatic change.

It is more effective a step back from channelization or a policy of removing small dams than the efforts to pass sediment at

large dams, in order to provide sand to the beaches at the delta.

Hydrology, Natural Hazards and Earth Systems research sometimes stress technological innovation in the field of data taking,

continuous monitoring and digital communication. However, documentary research (such as Llasat et al., 2005 and others)

and archival perusal in search of old maps and photographs should be encouraged as well. The fate of the Llobregat river could

not have been disclosed without that kind of work.

**Code/Data availability**

Not applicable

**Author contribution**

The second author did most of the basic research, oriented by the other two authors.

**Competing interests**

The authors declare that they have no conflict of interest.

**Acknowledgements**

Thanks to National Archives of Catalonia (ANC), Víctor Ténez, Vicenç Gràcia, Javier Martín-Vide and Carles Ibáñez.

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
