# Peer review of "What controls the coarse sediment yield to a Mediterranean delta The case of the Llobregat river (NE Iberian Peninsula)"

_Natural Hazards and Earth System Sciences, 2020_

## Referee Comment (RC1) · Carles Ibáñez (Referee) · 3 Apr 2020

Very interesting paper that, however, shows some important caveats that need to be solved before a decision of publication can be made. The paper needs a major change in the focus and the specific goals to sort out the weak points that contains right now. The starting point is the observed secular coastal retreat of the Llobregat Delta. This is an interesting new piece of information that gives value to the manuscript but at the same time shows the limitation of the approach taken. For me it was a surprise to see that the Delta was already retreating quickly by the end of the XIX century and that the retreat kept going all over the time till nowadays. This is very interesting, but in my opinion the possible causes argued in the manuscript to explain it are not convincing. In terms of damming the authors mention two dams built in the last decades in the upper basin, while river channelization is also relatively recent and cannot be the main cause of such a dramatic coastal erosion. The main argued possible cause is reforestation, but again the process is not so widespread from the beginning of the observed retreat and the increase in forest cover along the study period is not so large to explain the most of the deficit in sand delivery to the delta. According to Table 2 forest shifted from a cover of 63% in 1956 to a cover of 70% in 2009 for the whole river basin, and this is the main period of afforestation, mostly driven by the abandonment of traditional farming and public policies during the last decades of the dictatorship regime. At the same time, data also shows that large floods (a major source of sand delivery to the coast) have apparently been occurring all along the study period (a more detailed analysis of the changes in river floods along time could help to understand what's going on). There must be other causes to explain the sediment deficit in the delta, and the main one that comes to me is the widespread construction of weirs in the Llobregat River and its main tributaries (such as the Cardener) for industrial production (mostly textile) and for hydropower, that was already important in the XIX century. This chain of small reservoirs certainly modified in a dramatic way the hydro-sedimentary dynamics of the Llobregat River and tributaries, and could mostly explain what happened in the Llobregat Delta in terms of erosion. Thus, the paper needs to investigate this point as much as possible, both in terms of data (on the evolution of damming in the basin), mechanisms (how this damming modifies the sedimentary dynamics) and potential effects on sand delivery to the coast. In relation to the other analysed mechanisms that could explain in part the changes in river sediment dynamics and delivery to the coast (section 4), I have some other relevant comments: Land uses and urbanization: as mentioned in the text the change in forest cover is modest, I do not think it can be claimed as the main reason for the sediment deficit in the delta, though it may have some effect (see Ibáñez et al. 2019 and Nienhuis et al. 2020). Besides analysing the changes in land use, is there any possibility to estimate the relative contribution of this phenomenon to the sediment

deficit? (the same question applies to the other drivers of change in sediment dynamics in the river). Dams (sediment trapping): the authors mention that the percent of sediment retention in the two reservoirs of the upper basin may be proportional to the percent surface area that they close. However, it is well known that most of the erosion worldwide comes from the upper parts of the river basins. See for instance Wilkinson & McElroy (2007): Consideration of the variation in large river sediment loads and the geomorphology of respective river basin catchments suggests that natural erosion is primarily confined to drainage headwaters; âĹij83% of the global river sediment flux is derived from the highest 10% of Earth's surface. Then one should expect a higher proportion of sediment retention due to the two dams, which would be concentrated in the last decades, after dam construction. Dams (hydrological changes): I am not sure it's a good idea to combine the effect of dam regulation with river engineering to estimate changes in sediment delivery to the coast. In any case, it would be important to have at least an estimate of the change in carrying capacity for the whole river, not only the lower basin. Climate change (rainfall and runoff): this possible driver of change in sediment delivery to the coast has been neglected and could be significant. Sand transport capacity is mostly driven by river flow, so changes in river flow due to changes in rainfall and runoff could play a significant role. This possibility should be analysed (see Xing et al., 2014). Channelization and flood plain alteration (river engineering): again the analysis of the alteration of the river bed and the alluvial valley focuses only in the lower river basin, but is quite clear that most of the river basin is engineered (including small dams and other works). So, what is the global contribution of river engineering to the reduction of sediment delivery to the delta? Please try to make a global estimate if possible. Sand mining: it is mentioned but it would be interesting to have more quantitative information to know the relevance of this activity on the sand deficit to the delta. Other relevant comments regarding beach retreat (section 3): It would be interesting to add an extra graph or table to assess the evolution of the coastal erosion in the delta all over the study period, for instance in the river mouth, in order to see if there is any trend along time and also try to see if this trends match

with the assessed trends in sediment delivery to the coast. Sediment dynamics in the delta: "The reach is a sedimentary unit throughout the whole period 1891-1981" (lines 60-61). This is not strictly correct, depends on the interpretation of the sentence. Figure 3 shows two different sedimentary units "erosion-accretion" (quite typical in many deltas). This is likely explained by the existence of an old river mouth around Km 10. So it is a sedimentary unit composed by two sub-units. Limits of the Llobregat Delta and sand losses Southwards: "An oval contour slightly protruding into the sea, geographically speaking the delta, can be assigned to the length between x=15 and x=24 km, being the river mouth at x= 21 km" (lines 73-74). "The calculation yields a deficit of 57.000 m3/yr in the delta (x= 15-24 km) and a surplus of 29.000 m3/yr in the beaches west of it (X= 0-15 km)" (lines 93-94). The two sentences should be modified, since the delta is the whole stretch from km 0 to km 24. All deltas have sections with erosion and the corresponding sections with accretion due to the eroding stretch located "upstream" (in relation to the long-shore transport). "The negative balance (loss of sand) can be explained by the partially open western boundary (at x=0)" (lines 101-102). I am no sure that this is the correct explanation. Is there information showing that this volume of sand leaving the delta (quite a lot) is accumulating nearby? Could be the case that there are errors in the calculation of the sediment budget?

Last but not least I recommend to change the structure and title of the manuscript. I suggest something like: "Changes in coarse sediment delivery to the coast during the last century in the Llobregat River: causes and consequences". In terms of structure I would simplify it and present data in a more integrated way, including a table summarizing the estimated contribution of each component to the changes in sediment delivery and what are the data gaps necessary to get a better estimate.

References Ibáñez, C., Alcaraz, C., Caiola, N., Prado, P., Trobajo, R., Benito, X., ... & Syvitski, J. P. M. (2019). Basin-scale land use impacts on world deltas: Human vs natural forcings. Global and planetary change, 173, 24-32. Nienhuis, J. H., Ashton, A. D., Edmonds, D. A., Hoitink, A. J. F., Kettner, A. J., Rowland, J. C., & Törnqvist, T. E.

(2020). Global-scale human impact on delta morphology has led to net land area gain. Nature, 577(7791), 514-518. Wilkinson, B. H., & McElroy, B. J. (2007). The impact of humans on continental erosion and sedimentation. Geological Society of America Bulletin, 119(1-2), 140-156. Xing, F., Kettner, A. J., Ashton, A., Giosan, L., Ibáñez, C., & Kaplan, J. O. (2014). Fluvial response to climate variations and anthropogenic perturbations for the Ebro River, Spain in the last 4000 years. Science of the total environment, 473, 20-31.

---

## Referee Comment (RC2) · Anonymous Referee #2 · 16 May 2020

**Review of paper nhess-2020-72**

What controls the coarse sediment yield to a Mediterranean delta. The case of the
Llobregat river (NE Iberian Peninsula)

16$^{\text{th}}$ May, 2020

The authors study the importance of several factors in the sediment yield of the lower Llobregat River. The main finding is that flow regulation due to dam construction, and the construction of infrastructure in the floodplain, are the main factors explaining the change in sediment yield in the last decades. The manuscript presents a broad historical overview of all interventions and changes in the basin, as well as a comprehensive data set. Data from different sources is compiled, which makes the manuscript of interest for all future studies about the Llobregat River. The less strong point regards the estimation of sediment transport. From my point of view, the authors could have used some other tools that would allow a more precise estimation. Overall, the manuscript is interesting for the scientific community as well as for practitioners and certainly contributes to the understanding of the dynamics of the Llobregat River. Hence, I would recommend its publication, possibly after considering the comments below.

There are some major points to be discussed. The authors estimate the change in supply of bed load by measuring the change in alluvial surface area (Section 9). They also estimate the change in bed load carrying capacity (Section 10) and study the difference between these two parameters for predicting change in bed elevation. This would not be the approach I would follow, as a local reduction of the alluvial area does not indicate a decrease in the sediment supply at the same location. The sediment yield is indeed dependent on the basin area (studied in Section 5), but not on the floodplain area. A reduction of the width at a certain location does not decrease the load at this location, also not immediately downstream. A local reduction of the river width, and subsequent decrease in alluvial area, would, in general, cause an increase in the flow velocity (and bed shear stress), which would cause an increase of the sediment transport rate at the narrowed section, which implies an increase of the sediment yield to the downstream sections. Thus, a reduction of the river width causes short term aggradation downstream of the narrowed reach, as well as degradation at the upstream end of the narrowed reach. *Jansen et al.* (1979) provides a general overview of the effect of such interventions.

In estimating changes in bed elevation, one could simply apply mass balance (i.e., the *Exner* (1920) equation). Gradients in sediment transport rate lead to changes in bed elevation. For instance, in Table 5, one find that the sediment transport capacity of Reach 2 is more than twice the sediment transport capacity of Reach 1. Assuming, as the authors mention, that locally there is no lack of alluvial sediment (i.e., there is no bed rock or non-erodible layer), one would expect degradation to have occurred.

The paper is mainly about data analysis. Nevertheless, when estimating sediment transport and changes in bed elevation, I would have proposed to use a standard one-dimensional numerical model coupling the *Saint-Venant* (1871) equations to the *Exner* (1920) equation. Although the authors seem to have data for building a model with the actual geometry, a simplified model with schematic cross-sections would suffice to estimate orders of magnitude. This approach would, for instance, prevent the crude limitation of assuming that changes propagate from one reach to the next one (with different lengths) in the time in which data is available (which is not constant). A second benefit would be the capacity of assessing certainty in the results, as one could easily estimate the load using different sediment transport relations and grain size. Moreover, it would allow to estimate the change in bed elevation due to particular flood events, which is currently not possible to estimate, although the authors mention that are very relevant.

As a minor point, I would highlight that the manuscript reads well, but closer to a book than to a research article. The manuscript is organized in 15 sections, including an epilogue. I find the data presented in the epilogue to be as important (probably even more) as the data presented in previous

sections and makes me wonder why it is considered an epilogue. Although a matter of taste, some readers may find clearer to have the usual structure with methodology and data sources, results, discussion, and conclusions.

At some locations in the paper, I missed references. For locals, the sources may be obvious, but international readers would benefit from a reference to the sentences in lines 49 and 50 where it is mentioned that "archaeologist say" and "geologist say". Similarly, the authors mention that the travel time of a flood wave is 22 h and a reference would help the reader. In Line 125 the authors mentions that "land use changes (...) have been analysed with the best aerial photographs (...) and a modern land use map". I would like to know more about the analysis itself. Please also check references. I could not find the reference in Line 455 in the text.

A last minor point is about the use of tables. While easier for reading the exact values, in my view trends are more easily perceived from plots. Also, please consider Table 1, where the reader finds surplus and deficit with signs. Possibly a bar plot may be a good way to present the results. Please also consider adding the figure about the duration curves (Line 178).

Concluding, the manuscript is interesting for the community and in my view it deserves publication. A stronger modelling would substantially increase the value of the work done, but I understand that the focus is on data analysis. A different structure may help certain readers, although this is more of personal taste than an objective assessment.

**References**

Exner, F. M. (1920), Zur Physik der Dünen, *Akad. Wiss. Wien Math. Naturwiss*, *129*(2a), 929–952, (in German).

Jansen, P. P., L. Van Bendegom, J. Van den Berg, M. De Vries, and A. Zanen (1979), *Principles of river engineering: the non-tidal alluvial river*, 509 pp., Pitman London.

Saint-Venant, A. J. C. B. (1871), Théorie du mouvement non permanent des eaux, avec application aux crues des rivières et à l'introduction des marées dans leur lit, *Comptes Rendus des séances de l'Académie des Sciences*, *73*, 237–240, (in French).

---

## Author Comment (AC1) · 19 May 2020

ANSWER TO Review nhess-2020-72, Martín-Vide et al., by Carles Ibáñez

REVIEWER:Very interesting paper that, however, shows some important caveats that need to be solved before a decision of publication can be made. The paper needs a major change in the focus and the specific goals to sort out the weak points that contains right now. The starting point is the observed secular coastal retreat of the Llobregat Delta. This is an interesting new piece of information that gives value to the manuscript but at the same time shows the limitation of the approach taken. For me it was a surprise to see that the Delta was already retreating quickly by the end of

the XIX century and that the retreat kept going all over the time till nowadays. This is very interesting, but in my opinion the possible causes argued in the manuscript to explain it are not convincing. In terms of damming the authors mention two dams built in the last decades in the upper basin, while river channelization is also relatively recent and cannot be the main cause of such a dramatic coastal erosion. The main argued possible cause is reforestation, but again the process is not so widespread from the beginning of the observed retreat and the increase in forest cover along the study period is not so large to explain the most of the deficit in sand delivery to the delta. According to Table 2 forest shifted from a cover of 63% in 1956 to a cover of 70% in 2009 for the whole river basin, and this is the main period of afforestation, mostly driven by the abandonment of traditional farming and public policies during the last decades of the dictatorship regime. At the same time, data also shows that large floods (a major source of sand delivery to the coast) have apparently been occurring all along the study period (a more detailed analysis of the changes in river floods along time could help to understand what's going on).

AUTHORS:The reviewer is right. The land use change is now considered a minor factor. The role of the floods has deserved more consideration in the new text, by evaluation of the floods in the XX century, as well as in the XIX century.

REVIEWER: There must be other causes to explain the sediment deficit in the delta, and the main one that comes to me is the widespread construction of weirs in the Llobregat River and its main tributaries (such as the Cardener) for industrial production (mostly textile) and for hydropower, that was already important in the XIX century. This chain of small reservoirs certainly modified in a dramatic way the hydro-sedimentary dynamics of the Llobregat River and tributaries, and could mostly explain what happened in the Llobregat Delta in terms of erosion. Thus, the paper needs to investigate this point as much as possible, both in terms of data (on the evolution of damming in the basin), mechanisms (how this damming modifies the sedimentary dynamics) and potential effects on sand delivery to the coast.

AUTHORS: The reviewer is right. Thanks for the suggestion, which has produced a strong change in the discussion section of the new text. It has been proved the paramount role of these weirs (small dams), standing in the middle reaches of the river, on the sediment dynamics. Parts of the new discussion are:

[revised manuscript text omitted]

REVIEWER In relation to the other analysed mechanisms that could explain in part the changes in river sediment dynamics and delivery to the coast (section 4), I have some

other relevant comments: Land uses and urbanization: as mentioned in the text the change in forest cover is modest, I do not think it can be claimed as the main reason for the sediment deficit in the delta, though it may have some effect (see Ibáñez et al. 2019 and Nienhuis et al. 2020). Besides analysing the changes in land use, is there any possibility to estimate the relative contribution of this phenomenon to the sediment deficit? (the same question applies to the other drivers of change in sediment dynamics in the river).

AUTHORS: The effect of changes in land use are rather connected to the wash load component of the sediment transport, which was not the purpose of the paper (it was bed load instead). We could not have improved the estimates done in the references mentioned by the reviewer, which are duly incorporated in the new paper.

REVIEWER: Dams (sediment trapping): the authors mention that the percent of sediment retention in the two reservoirs of the upper basin may be proportional to the percent surface area that they close. However, it is well known that most of the erosion worldwide comes from the upper parts of the river basins. See for instance Wilkinson & McElroy (2007): Consideration of the variation in large river sediment loads and the geomorphology of respective river basin catchments suggests that natural erosion is primarily confined to drainage headwaters; âĹij83% of the global river sediment flux is derived from the highest 10% of Earth's surface. Then one should expect a higher proportion of sediment retention due to the two dams, which would be concentrated in the last decades, after dam construction.

AUTHORS: The reviewer is right. The paragraph about sediment trap in reservoirs is corrected accordingly, with this argument:

Sediment load coming from the regulated basin as wash load will be mostly trapped in the reservoirs, but the wash load component of the sediment yield, having grainsizes in the clay-silt range (up to 62 $\mu$m), is not relevant for the coastline evolution, made of fine sand (280 $\mu$m). Regarding the load coming from the channels, ultimately trapped

in reservoirs, the drainage network density is similar all over the whole basin, but main rivers and tributaries are steeper in the mountainous regulated basins. Thus, the supply of coarse sediment from channels to the reservoirs is probably larger than 16.5% of the same load at the river mouth (Wilkinson and McElroy, 2007). Sediment supply is resumed in §9.

REVIEWER: Dams (hydrological changes): I am not sure it's a good idea to combine the effect of dam regulation with river engineering to estimate changes in sediment delivery to the coast. In any case, it would be important to have at least an estimate of the change in carrying capacity for the whole river, not only the lower basin.

AUTHORS: The combination of dam regulation and river works produced an estimate of the reduction in carrying capacity in the lower Llobregat, as the reviewer says. This was based on careful archival research of historical river cross-sections, bed gradients and so on, which was possible for the very populated lower Llobregat, not anywhere else. However, the same reduction in carrying capacity for the rest of the river is attempted in the discussion just thanks to the new information about the weirs (small dams).

REVIEWER: Climate change (rainfall and runoff): this possible driver of change in sediment delivery to the coast has been neglected and could be significant. Sand transport capacity is mostly driven by river flow, so changes in river flow due to changes in rainfall and runoff could play a significant role. This possibility should be analysed (see Xing et al., 2014).

AUTHORS: Climate change has not been neglected, actually, but put aside. We think climate change is essential to predict the future of the river and delta, but the purpose of the paper was to explain the past changes, specially in the main period of analysis 1946-1981 (in the discussion it has been extended back to XIX century).

REVIEWER: Channelization and flood plain alteration (river engineering): again the analysis of the alteration of the river bed and the alluvial valley focuses only in the

lower river basin, but is quite clear that most of the river basin is engineered (including small dams and other works). So, what is the global contribution of river engineering to the reduction of sediment delivery to the delta? Please try to make a global estimate if possible.

AUTHORS: The role of river engineering in most of the river is now dealt with in the discussion. It is clear now the paramount role of these weirs (small dams), which sum up almost 400 m of head and many kilometres of influence. The estimate we make now is that 80% of the reduction of sediment delivery to the delta is due to the XIX-century engineering works in the middle reaches (small dams) IN COMBINATION WITH a period of anomalous hydrology in 1830-1870. The remaining 20% is due to the XX century encroachment by infrastructures.

REVIEWER: Sand mining: it is mentioned but it would be interesting to have more quantitative information to know the relevance of this activity on the sand deficit to the delta.

AUTHORS: Although there are some data about mining, it is not clear to what extent these extractions participate in the bed load dynamics, provided that most of them are located in the floodplains.

REVIEWER: Other relevant comments regarding beach retreat (sect ion 3): It would be interesting to add an extra graph or table to assess the evolution of the coastal erosion in the delta all over the study period, for instance in the river mouth, in order to see if there is any trend along time and also try to see if this trends match with the assessed trends in sediment delivery to the coast.

AUTHORS: This is done in table 8, by doing our best with the data. For the river mouth specifically, the graph in fig.3 is what the reviewer is asking for. Note that the comparison between the delta and the sediment delivery is restricted to long periods of roughly one decade, because of the calendar of aerial photographs at those times.

REVIEWER: Sediment dynamics in the delta: "The reach is a sedimentary unit throughout the whole period 1891-1981" (lines 60-61). This is not strictly correct, depends on the interpretation of the sentence. Figure 3 shows two different sedimentary units "erosion-accretion" (quite typical in many deltas). This is likely explained by the existence of an old river mouth around Km 10. So it is a sedimentary unit composed by two sub-units.

AUTHORS: The reviewer is right. The text is changed accordingly.

Limits of the Llobregat Delta and sand losses Southwards: "An oval contour slightly protruding into the sea, geographically speaking the delta, can be assigned to the length between x=15 and x=24 km, being the river mouth at x= 21 km" (lines 73-74). "The calculation yields a deficit of 57.000 m3/yr in the delta (x= 15-24 km) and a surplus of 29.000 m3/yr in the beaches west of it (X= 0-15 km)" (lines 93-94). The two sentences should be modified, since the delta is the whole stretch from km 0 to km 24. All deltas have sections with erosion and the corresponding sections with accretion due to the eroding stretch located "upstream" (in relation to the long-shore transport).

AUTHORS: The reviewer is right, also. The text is changed accordingly.

REVIEWER: "The negative balance (loss of sand) can be explained by the partially open western boundary (at x=0)" (lines 101-102). I am no sure that this is the correct explanation. Is there information showing that this volume of sand leaving the delta (quite a lot) is accumulating nearby? Could be the case that there are errors in the calculation of the sediment budget?

AUTHORS: The error in the budget, computed on the grounds of aerial photographs and the USGS procedure, must be small, not more than the accuracy of aerial photographs. We could not find information showing this loss of sand, not even in the publications by well-known CIIRC research centre (referred in the paper).

REVIEWER: Last but not least I recommend to change the structure and title of the

manuscript. I suggest something like: "Changes in coarse sediment delivery to the coast during the last century in the Llobregat River: causes and consequences".

AUTHORS: We remained attached to our title. We think that "What controls" stands for "Changes in" in the reviewer's alternative title, ídem "yield" for "delivery", "to a delta" for "to the coast". The time reference "during the last century" in the alternative is misleading in our view, because the centre of attention is only 1946-1981 and secondly because we have had to look back to 1816 (date of the starting of factory construction), that is to say to TWO centuries. But a title "...to the coast during the last TWO centuries. . ." would also be misleading, in our view.

REVIEWER: In terms of structure I would simplify it and present data in a more integrated way, including a table summarizing the estimated contribution of each component to the changes in sediment delivery and what are the data gaps necessary to get a better estimate.

AUTHORS: The table summarizing the contribution of each component is well beyond our knowledge and abilities. However, some 80% share by the old engineering works in the middle river, and the remaining 20% by the new encroachment of the lower river, was obtained and is highlighted in the conclusion section.

More specifically, this argument goes this way:

The computed annual river yield in 1946-1981 from $\approx 16\times103$ to $\approx 10\times103$ m3/yr is found to be a substantial factor for the delta evolution. It is of the same order of magnitude but lower than the delta balance (―$28\times103$ m3/yr). Its variation of $\approx$―$6\times103$ m3/yr between 1956 to 1981 due to the river encroachment by infrastructures, which is our main research objective, is less substantial, but it still accounts for $\approx 20\%$ of the balance. The role of the regulation by dams, $\approx$ ―$3.5\times103$ m3/yr, accounts for some 12% of the balance. It must be recalled that the computation is based upon normal flows and annuals floods, not including large floods, whereas the delta evolution (§2) encompasses all phenomena.

REVIEWER: I recommend major changes References Ibáñez, C., Alcaraz, C., Caiola, N., Prado, P., Trobajo, R., Benito, X., ... & Syvitski, J. P. M. (2019). Basin-scale land use impacts on world deltas: Human vs natural forcings. Global and planetary change, 173, 24-32. Nienhuis, J. H., Ashton, A. D., Edmonds, D. A., Hoitink, A. J. F., Kettner, A. J., Rowland, J. C., & Törnqvist, T. E. (2020). Global-scale human impact on delta morphology has led to net land area gain. Nature, 577(7791), 514-518. Wilkinson, B. H., & McElroy, B. J. (2007). The impact of humans on continental erosion and sedimentation. Geological Society of America Bulletin, 119(1-2), 140-156. Xing, F., Kettner, A. J., Ashton, A., Giosan, L., Ibáñez, C., & Kaplan, J. O. (2014). Fluvial response to climate variations and anthropogenic perturbations for the Ebro River, Spain in the last 4000 years. Science of the total environment, 473, 20-31.

AUTHORS: Most the theses references have been mentioned/discussed in the text and added to the reference list. More specifically, these are:

Alayo, J.C. Water and Energy. Hydropower in the Catalan rivers (in Catalan), Pagès ed., Barcelona, 936pp, 2017.

Barriendos, M., Rodrigo, F.S. Study of historical flood events on Spanish rivers using documentary data. Hydrological Sciences Journal, 51(5), 765-783, 2006.

Barriendos, M., Gil-Guirado, S., Pino, D., Tuset, J., Pérez-Morales, A., Alberola, A., Costa, J., Balasch, J.C., Castelltort, X., Mazón, J., Ruiz-Bellet, J.Ll. Climatic and social factors behind the Spanish Mediterranean flood event chronologies from documentary sources (14th-20th centuries). Global and Planetary Change 182, 102997, 2019.

Ibáñez, C., Alcaraz, C., Caiola, N., Prado, P., Trobajo, R., Benito, X., Day, J.W., Reyes, E., Syvitski, J. P. M. Basin-scale land use impacts on world deltas: Human vs natural forcings. Global and planetary change, 173, 24-32, 2019.

Llasat, M.C., Barriendos, M., Barrera, A., Rigo, T. Floods in Catalonia (NE Spain) since the 14th century. Climatological and meteorological aspects from historical documentary sources and old instrumental records. J. of Hydrology 313, 32-27, 2005.

Marcos Valiente, O. Recent changes in the Llobregat delta coastline (in Catalan). I.E.C., 1995.

Marquès, M. A. The Quaternary of the Llobregat delta (in Catalan). Inst. d'Estudis Catalans. 208 pp, Barcelona, 1984.

Paladella, F., Faura Sans, M. Experiences about the Llobregat delta progradation. (in Catalan). Arxius de l'Escola Superior d'Agricultura", 1935. https://upcommons.upc.edu/handle/2099/11137.

Wilkinson, B. H., McElroy, B. J. The impact of humans on continental erosion and sedimentation. Geological Society of America Bulletin, 119(1-2), 140-156, 2007.

Xing, F., Kettner, A. J., Ashton, A., Giosan, L., Ibáñez, C., Kaplan, J. O. Fluvial response to climate variations and anthropogenic perturbations for the Ebro River, Spain in the last 4000 years. Science of the total environment, 473, 20-31, 2014.
* * *
[Figure]

**Fig. 1.** Cumulative height H (m) versus calendar date for the installation of factories in the middle reaches of the river (data in Alayo, 2017), and its effects at the upper border of the lower Llobregat, und

[Figure]

[Figure]

**Fig. 2.** Bridge close to town D, shot by well-known French photographer Jean Laurent probably in 1866-1867. The only bridge in lower Llobregat at that time had a total length 334,36 metres, with 15 arches, the

---

## Author Response (AR1)

RESPONSE TO REVIEW by Carles Ibáñez. Our answers in blue

Very interesting paper that, however, shows some important caveats that need to be solved before a decision of publication can be made. The paper needs a major change in the focus and the specific goals to sort out the weak points that contains right now.

5  The starting point is the observed secular coastal retreat of the Llobregat Delta. This is an interesting new piece of information that gives value to the manuscript but at the same time shows the limitation of the approach taken. For me it was a surprise to see that the Delta was already retreating quickly by the end of the XIX century and that the retreat kept going all over the time till nowadays. This is very interesting, but in my opinion the possible causes argued in the manuscript to explain it are not convincing. In terms of damming the authors

10  mention two dams built in the last decades in the upper basin, while river channelization is also relatively recent and cannot be the main cause of such a dramatic coastal erosion. The main argued possible cause is reforestation, but again the process is not so widespread from the beginning of the observed retreat and the increase in forest cover along the study period is not so large to explain the most of the deficit in sand delivery to the delta. According to Table 2 forest shifted from a cover of 63% in 1956 to a cover of 70% in 2009 for the whole river

15  basin, and this is the main period of afforestation, mostly driven by the abandonment of traditional farming and public policies during the last decades of the dictatorship regime. At the same time, data also shows that large floods (a major source of sand delivery to the coast) have apparently been occurring all along the study period (a more detailed analysis of the changes in river floods along time could help to understand what's going on).

The reviewer is right. The land use change is now considered a minor factor. The role of the floods has deserved

20  more consideration in the new text.

There must be other causes to explain the sediment deficit in the delta, and the main one that comes to me is the widespread construction of weirs in the Llobregat River and its main tributaries (such as the Cardener) for industrial production (mostly textile) and for hydropower, that was already important in the XIX century. This chain of small reservoirs certainly modified in a dramatic way the hydro-sedimentary dynamics of the Llobregat

25  River and tributaries, and could mostly explain what happened in the Llobregat Delta in terms of erosion. Thus, the paper needs to investigate this point as much as possible, both in terms of data (on the evolution of damming in the basin), mechanisms (how this damming modifies the sedimentary dynamics) and potential effects on sand delivery to the coast.

The reviewer is right. Thanks for the suggestion, which has produced a strong change in the discussion section of

30  the new text. It has been proved the paramount role of these weirs (small dams), standing in the middle reaches of the river, on the sediment dynamics.

In relation to the other analysed mechanisms that could explain in part the changes in river sediment dynamics and delivery to the coast (section 4), I have some other relevant comments:

Land uses and urbanization: as mentioned in the text the change in forest cover is modest, I do not think it can

35  be claimed as the main reason for the sediment deficit in the delta, though it may have some effect (see Ibáñez et al. 2019 and Nienhuis et al. 2020). Besides analysing the changes in land use, is there any possibility to estimate

the relative contribution of this phenomenon to the sediment deficit? (the same question applies to the other drivers of change in sediment dynamics in the river).

The effect of changes in land use are rather connected to the wash load component of the sediment transport, which was not the purpose of the paper (it was bed load instead). We could not have improved the estimates done in the references mentioned by the reviewer, which are duly incorporated in the new paper.

Dams (sediment trapping): the authors mention that the percent of sediment retention in the two reservoirs of the upper basin may be proportional to the percent surface area that they close. However, it is well known that most of the erosion worldwide comes from the upper parts of the river basins. See for instance Wilkinson & McElroy (2007): Consideration of the variation in large river sediment loads and the geomorphology of respective river basin catchments suggests that natural erosion is primarily confined to drainage headwaters; ~83% of the global river sediment flux is derived from the highest 10% of Earth's surface.

Then one should expect a higher proportion of sediment retention due to the two dams, which would be concentrated in the last decades, after dam construction.

The reviewer is right. The paragraph about sediment trap in reservoirs is corrected accordingly.

Dams (hydrological changes): I am not sure it's a good idea to combine the effect of dam regulation with river engineering to estimate changes in sediment delivery to the coast. In any case, it would be important to have at least an estimate of the change in carrying capacity for the whole river, not only the lower basin.

The combination of dam regulation and river works produced an estimate of the reduction in carrying capacity in the lower Llobregat, as the reviewer says. This was based on careful archival research of historical river cross-sections, bed gradients and so on, which was possible for the very populated lower Llobregat, not anywhere else. However, the same reduction in carrying capacity for the rest of the river is attempted in the discussion just thanks to the new information about the weirs (small dams).

Climate change (rainfall and runoff): this possible driver of change in sediment delivery to the coast has been neglected and could be significant. Sand transport capacity is mostly driven by river flow, so changes in river flow due to changes in rainfall and runoff could play a significant role. This possibility should be analysed (see Xing et al., 2014).

Climate change has not been neglected, actually, but put aside. We think climate change is essential to predict the future of the river and delta, but the purpose of the paper was to explain the past changes, specially in the main period of analysis 1946-1981 (in the discussion it has been extended back to XIX century).

Channelization and flood plain alteration (river engineering): again the analysis of the alteration of the river bed and the alluvial valley focuses only in the lower river basin, but is quite clear that most of the river basin is engineered (including small dams and other works). So, what is the global contribution of river engineering to the reduction of sediment delivery to the delta? Please try to make a global estimate if possible.

70    The role of river engineering in most of the river is now dealt with in the discussion. It is clear now the paramount role of these weirs (small dams), which sum up almost 400 m of head and many kilometres of influence. The estimate we make now is that 80% of the reduction of sediment delivery to the delta is due to the XIX-century engineering works in the middle reaches (small dams) IN COMBINATION WITH a period of anomalous hydrology in 1830-1870. The remaining 20% is due to the XX century encroachment by infrastructures.

75    Sand mining: it is mentioned but it would be interesting to have more quantitative information to know the relevance of this activity on the sand deficit to the delta.

Although there are some data about mining, it is not clear to what extent these extractions participate in the bed load dynamics, provided that most of them are located in the floodplains.

Other relevant comments regarding beach retreat (sect ion 3):

80    It would be interesting to add an extra graph or table to assess the evolution of the coastal erosion in the delta all over the study period, for instance in the river mouth, in order to see if there is any trend along time and also try to see if this trends match with the assessed trends in sediment delivery to the coast.

This is done in table 8, by doing our best with the data. For the river mouth specifically, the graph in fig.3 is what the reviewer is asking for. Note that the comparison between the delta and the sediment delivery is restricted to

85    long periods of roughly one decade, because of the calendar of aerial photographs at those times.

Sediment dynamics in the delta:

"The reach is a sedimentary unit throughout the whole period 1891-1981" (lines 60-61). This is not strictly correct, depends on the interpretation of the sentence. Figure 3 shows two different sedimentary units "erosion-accretion" (quite typical in many deltas). This is likely explained by the existence of an old river mouth around Km

90    10. So it is a sedimentary unit composed by two sub-units.

The reviewer is right. The text is changed accordingly.

Limits of the Llobregat Delta and sand losses Southwards:

"An oval contour slightly protruding into the sea, geographically speaking the delta, can be assigned to the length between x=15 and x=24 km, being the river mouth at x= 21 km" (lines 73-74). "The calculation yields a deficit of

95    57.000 m3/yr in the delta (x= 15-24 km) and a surplus of 29.000 m3/yr in the beaches west of it (X= 0-15 km)" (lines 93-94).

The two sentences should be modified, since the delta is the whole stretch from km 0 to km 24. All deltas have sections with erosion and the corresponding sections with accretion due to the eroding stretch located "upstream" (in relation to the long-shore transport).

100   The reviewer is right, also. The text is changed accordingly.

"The negative balance (loss of sand) can be explained by the partially open western boundary (at x=0)" (lines 101-102).

I am no sure that this is the correct explanation. Is there information showing that this volume of sand leaving the delta (quite a lot) is accumulating nearby? Could be the case that there are errors in the calculation of the sediment budget?

The error in the budget, computed on the grounds of aerial photographs and the USGS procedure, must be small, not more than the accuracy of aerial photographs. We could not find information showing this loss of sand, not even in the publications by well-known CIIRC research centre (referred in the paper).

Last but not least I recommend to change the structure and title of the manuscript. I suggest something like: "Changes in coarse sediment delivery to the coast during the last century in the Llobregat River: causes and consequences".

We remained attached to our title. We think that "What controls" stands for "Changes in" in the reviewer's alternative title, ídem "yield" for "delivery", "to a delta" for "to the coast". The time reference "during the last century" in the alternative is misleading in our view, because the centre of attention is only 1946-1981 and secondly because we have had to look back to 1816 (date of the starting of factory construction), that is to say to TWO centuries. But a title "…to the coast during the last TWO centuries…" would also be misleading, in our view.

In terms of structure I would simplify it and present data in a more integrated way, including a table summarizing the estimated contribution of each component to the changes in sediment delivery and what are the data gaps necessary to get a better estimate.

The table summarizing the contribution of each component is well beyond our knowledge and abilities. However, some 80% share by the old engineering works in the middle river, and the remaining 20% by the new encroachment of the lower river, was obtained and is highlighted in the conclusion section.

Most the theses references have been mentioned/discussed in the text and added to the reference list.

**RESPONSE TO ANONYMOUS REVIEW**

The authors thank the reviewer for his/her careful reading of the manuscript and fruitful comments. The paper is indeed, as reviewer says, a historical overview of the interventions and changes in the Llobregat basin and main channel (including some other interventions in the middle reach to be added to the final manuscript to be submitted, thanks to the first reviewer). What is probably a nice but excessive statement by the reviewer is that we have collected "a comprehensive data set", since we just were able, at great pains, to draw one typical cross-section for each of the five reaches of the lower Llobregat, together with one bed slope and one mean grain size, for each of the five dates analysed (from 1946 till 1981). If geometrical data of this kind having more resolution are certainly available for the last date (1981), the need of a fair comparison with the old dates warranted not to involve more detailed information from recent dates in the analysis.

The reviewer is right in the **fourth paragraph** in that it would have been better to use a one-dimensional model coupling Saint-Venant and Exner equations for the estimation of sediment transport and bed changes. Thanks to the comment, we plan to do that in the future. At the moment, however, we have two answers that may nuance the reviewer's assertion (they do nuance it in our view):

1) the very simple geometrical information available, mentioned in the paragraph above, makes less interesting such a numerical model (only five different cross-section, one per reach, along 30 km). It is so because the use of a mass of model results for comparison between different dates would require to average them very much in time (one decade) and space (several km), in a way which is not unequivocal; we wonder if it is not better, in general terms, to average the data before computation that to average the results after computation; in fact, we think that this is a topic of research in itself, to carry on in the future,

2) any model of the type mentioned should use an empirical bedload equation as "closure" of the system of Saint-Venant and Exner equations, that is to say a particular function for the unit bedload rate $q_s$ in Exner eq., no matter this being Meyer-Peter and Müller (as in the paper) or any other else. We think that, in this way, the role of a supply-controlled sediment transport, in the reaches where this is the case, is not captured in the model, which in our view is, therefore, a capacity sediment transport model, strictly speaking.

Nevertheless, we agree with the reviewer that such a model would at least improve the analysis in unsteady flow (although flood hydrographs of past events are rare, let alone any sediment transport rates in floods) and would also overcome the "crude" assumption of propagation of changes between reaches at the time of the available data. Regarding this point, the literature supports the figure used in the paper, 500 m/year in average. Furthermore, the interventions in the middle reach (to be submitted in the proposed final paper) shed more light in the sense that this figure may be reasonable.

We agree with the reviewer's reasoning in the **second paragraph**. This is how the river bed reacts to a local (or not so local) narrowing (for ex. a channelization), no doubt, as has been proved by experiments and by conceptual models of equilibrium, as well. However, again we express some nuances to the application of the reviewer's

point of view to our case, without questioning at all the physics in it. Is the narrowing paradigm appropriate to our case?:

1) the floodplain area, made of coarse sediment exposed to entrainment in case of overbank flows, more than the basin area, as reviewer claims, determines the bedload in the lower river, according to the concepts of sediment origin. In fact, in longer time lapses, longer reaches of the alluvial channel, further upstream of the lower Llobregat studied in the paper, would participate in providing coarse sediment for bedload transport to the delta (again, the proposed final paper deals with this).

2) it is never a pure narrowing, i.e. a width change in space, what occurs properly, but a cumbersome spatial-temporal width change, quite general for all reaches (though varied in intensity, it is true), from one date to the following ten years apart. Seen from the neighbour reach downstream, a reduction in the sediment supply has occurred, with one decade of time to have made it feel in its balance. Alternatively, seen from the neighbour reach upstream, a reduction in carrying capacity has occurred in the downstream reach, as well, because the width has reduced after a decade of time for bedload work. Both terms of the balance change. Finally, seen from the reach itself whose width has reduced, questions arise about whether this channel narrowing is externally imposed (as if in a channelization) or results from an upstream reduction in sediment supply or from a upstream-propagating incision process taking place on downstream reaches.

3) the reviewer's physical reasoning leads to a slope change within the narrowed reach in the long term (as the effect of narrowing develops). However, the channels slopes, taken from the archival sources of information, show very little changes in time. This point may make more clear that the physical approach of narrowing (reviewer's approach) and our approach of bed load supply and capacity averaged in long periods of time differ. Reality is only one, different approaches should converge to reality. Moreover, what reviewer points out in **third paragraph** is certainly right. Regarding this, we have not compared the capacities at the two contiguous reaches but one capacity at the reach downstream of the two and one supply from the one upstream of them, which is not necessarily equal to its capacity.

With respect to the rest of the review, we have included the references and details that the reviewer demands in the sixth paragraph. The section called "epilogue" is now section 12 entitled "The new mouth and closure of the computation with real data" (not epilogue any more). This section still stands after the computation, not in the data section, because the paper focus on the period 1946-1981, while the new mouth (2004) was a much later development of a very different nature. However, its role of allowing a check of the previous computation is highlighted with the words "closure… with real data". In table 1, the sign — for deficit and + for surplus, under the headings deficit and surplus, is redundant, but mistakes are avoided, in our view. We are attached to tables instead of bar plots, in spite of being more demanding for the reader. However, the flow duration curve is added to the new version of the paper.

THE REVISED VERSION IN TRACKED CHANGES OPTION FOLLOWS.

ALL CHANGES MADE IN THE REVISED VERSION WITH RESPECT TO THE FIRST ONE ARE VISIBLE IN THIS OPTION.

ANSWERS TO THE REVIEWERS, SPECIALLY WHERE WE HAVE REFERRED TO "new version to be
210    submitted" ARE MORE CLEAR THROUGH THIS OPTION.

215

220

225

230

[revised manuscript text omitted]
., a̶s̶ ̶f̶i̶g̶.̶7̶ ̶i̶l̶l̶u̶s̶t̶r̶a̶t̶e̶s̶,̶ it is affected by the depth increase, which implies an increase in shear stress. F̶o̶r̶ ̶h̶i̶g̶h̶ ̶d̶i̶s̶c̶h̶a̶r̶g̶e̶s̶,̶ ̶t̶h̶e̶ ̶u̶n̶i̶t̶ ̶s̶o̶l̶i̶d̶ ̶d̶i̶s̶c̶h̶a̶r̶g̶e̶ ̶o̶f̶ ̶M̶P̶M̶ ̶d̶e̶p̶e̶n̶d̶s̶ ̶r̶o̶u̶g̶h̶l̶y̶ o̶n̶ ̶d̶e̶p̶t̶h̶ ̶t̶o̶ ̶t̶h̶e̶ ̶3̶/̶2̶ ̶p̶o̶w̶e̶r̶.̶ ̶F̶o̶r̶ ̶t̶h̶e̶ ̶e̶x̶a̶m̶p̶l̶e̶ ̶i̶n̶ ̶f̶i̶g̶.̶7̶,̶ ̶t̶h̶i̶s̶ ̶m̶e̶a̶n̶s̶ ̶a̶ ̶s̶h̶e̶a̶r̶ ̶s̶t̶r̶e̶s̶s̶ (1.80 times higher in 2016 than in 1946 in the case of fig.8).,̶ ̶w̶h̶i̶c̶h̶ ̶a̶t̶t̶e̶n̶u̶a̶t̶e̶s̶ ̶t̶h̶e̶ ̶l̶a̶r̶g̶e̶ ̶d̶e̶c̶r̶e̶a̶s̶e̶ ̶i̶n̶ ̶a̶l̶l̶u̶v̶i̶a̶l̶ ̶w̶i̶d̶t̶h̶ ̶b̶e̶t̶w̶e̶e̶n̶ ̶t̶h̶e̶s̶e̶ ̶t̶w̶o̶ ̶d̶a̶t̶e̶s̶ ̶(̶f̶r̶o̶m̶ ̶1̶7̶5̶ ̶m̶ ̶t̶o̶ ̶3̶3̶ ̶m̶ ̶i̶n̶ ̶a̶v̶e̶r̶a̶g̶e̶ w̶i̶d̶t̶h̶,̶ ̶t̶a̶b̶l̶e̶ ̶4̶)̶.̶

[revised manuscript text omitted]

---

## Referee Report (RR1)

660

[referee-annotated manuscript omitted]

---

## Author Response (AR2)

**TRACKED CHANGES**

[revised manuscript text omitted]

> **Commented [IC1]:** The abstract does not reflect properly the content of the paper and is not well structures, it must be re-written. We made some changes

> **Commented [IC2]:** Are not all rivers alluvial? Some rivers run on cohesive material.

over which the main pressure has been one of channelization, yet some information prior to this period will be necessary to understand the long term trends. The practice of channelizing a river generally involves increasing channel capacity and so, an erosional response, due to an enhanced sediment carrying capacity, is to be feared, although this is not always the case (Simon and Rinaldi, 2006). Typically, it also involves narrowing of the flood channel by taking a large part of the floodplains out of the hydraulic conveyance system , under the pressure of urban sprawl. This floodplain width reduction (encroachment or contraction) implies a perturbation of the equilibrium (more specifically, a degradation), as demonstrated analytically and experimentally by Vanoni (1975), yet this is only one of the several causes of the degradation of a river bed (Galay, 1983).

As regards the delta, the relative importance of fluvial building and wave and tidal reworking determines the delta morphology and evolution (Bridge, 2003). The relevant maritime factors are reduced to wave action in the case of the Mediterranean sea (no substantial tides). This wave action and its related currents produce a certain longitudinal coastal sediment transport, as well as a transfer of sand towards the open sea. The dominance of the fluvial or the maritime factor varies in space and time for a given delta. However, the simple statement made herein is that the greater the river sediment supply the more the delta will protrude into the standing water body, to equality of the maritime factor, and vice versa. Literature on delta evolution is abundant (e.g. Orton and Reading, 1993, Syvitski and Saito, 2007) and on river evolution as well (e.g. Rinaldi and Simon, 1998, Martín-Vide et al, 2010), but the connection between the two is less well known in physical terms, in spite of statistical approaches (Ibañez et al., 2019, Xing et al. 2014). It is difficult to find data to evaluate the influence of sediment supply perturbations on delta evolution, except a few cases such as the Mississippi river (Allison et al, 2012, Viparelli et al, 2015). A connection of this type is attempted in this research.

The paper concentrates on the fluvial component, for which a method to compute the actual sediment transport at different decades is followed, by using real data on the long river profile, the grain size of the available alluvium and the annual high flows and small floods. The focus is on what controls the coarse sediment yield into the sea, nourishing the beaches with sand. The retreat of beaches (especially in deltas) is a big concern in the Mediterranean region ('coarse' means sand herein). What controls the yield into the sea implies, as a consequence, which measures are more sensible in order to keep providing enough sand to the beaches.

Llobregat River is 163 km-long and drains an area of 4925 km$^2$ of the Northeastern Iberian peninsula, with its headland in the Pyrenees mountain range (fig.1). Archeologists have found evidences of human presence in the delta since Roman times (Marquès, 1984). The present delta results from the Holocene transgression (6000 years ago), yet we are more interested in the delta evolution in the last century (within the so-call Anthropocene). The Latin name of the river was Rubricatus, which means dyed in red, in allusion to the color of its waters, probably because of its large fine sediment load. Moreover, Llobregat is a gravel-bed river upstream of its delta, with a high bed load transport capacity. The delta can be classified as sandy mixed

Commented [JP3]: done

Commented [IC4]: It is necessary a section of methods to explain in more detail the calculations, as it is now is no clear enough.

This is done as subsections in the appropriate paragraphs. In the introduction, we prefer not to do that. We opened 2 new methodological subsections, namely 2.1 and 10.1

Commented [IC5]: Is the word coarse adequated for sand? Yes, sand is classified in fine, medium and coarse

Commented [JP6]: done

Commented [IC7]: This sentence is not clear, please explain here the hypothesis, such as: we hypothesize that river channelization has been the main cause of coastal retreat in the Llobregat Delta.

This is more developed at the beginning of section 3. We think there, after having seen the coastal change, is more appropriate.

Commented [JP8]: done

[revised manuscript text omitted]

the river yield? Similar to what has been done about the beach retreat, we will primarily use historical information on the river condition in 1946-1981, although discussion of the results will require to go back to the river condition in the XIX century. Before that, the causes of decrease in river sediment yield are examined next.

**3 Causes of decrease in sediment yield**

The decrease of the sediment yield of a river to its delta may be due to different reasons. Here we will consider: a) land use changes including urbanization, b) the construction of small and large reservoirs, that 1) trap sediment and 2) regulate flow, and c) river engineering works of any kind (mining included) on the channel and floodplains.

Cause a) affects primarily one component of the sediment load, the wash load, i.e. the fine sediment coming from anywhere in the basin. Cause b) affects all components of the sediment load but certainly its coarse fraction, which is more prone to get trapped in reservoirs than wash load. Cause c) in the Llobregat case since 1946 there has been a progressive encroachment of the river by infrastructures (roads and railways) and its channelization against flooding with bank erosion control measures, in combination with some gravel and sand mining. These engineering works affect sediment load coming from the channel, composed of sand and gravel, not the wash load coming from the basin.

**4 Land uses and urbanization**

Land use changes in the Llobregat basin have been analyzed comparing the best aerial photographs from the past (1956) with a modern land use map (2009) (CREAF research center). The results are summarized in table 2, with aggregation of land uses in only three main categories: agriculture, forest and urban. The percentages for the whole Llobregat basin show a modest change consisting of a loss of agriculture land for the equitable benefit of towns (urban), on one side, and forest, which grow on the abandoned fields, on the other.

| | basin 4925 km$^2$ | | lower basin 343 km$^2$ | | tributary 3, 124 km$^2$ | |
|---|---|---|---|---|---|---|
| | 1956 | 2009 | 1956 | 2009 | 1956 | 2009 |
| agriculture | 35% | 22% | 43% | 8% | 45% | 9% |
| urban | 2% | 8% | 6% | 37% | 8% | 43% |
| forest | 63% | 70% | 51% | 55% | 47% | 48% |

Table 2. Land use change in the whole, lower basin and tributary 3 sub-basin in 1956-2009 (Prats-Puntí, 2018).

For the lower Llobregat basin, amounting 7% of the total basin area (fig.1), the loss of agricultural fields is more important and benefits more the urban area than the forest (fig.4). The lower Llobregat channel close to Barcelona is the most intervened reach. The case of the most urbanized sub-basin, the tributary 3 catchment (figs.1 and 4, table 2), shows a more marked reversal

**Commented [IC16]:** This should go in the previous section, as part of the goals and hypothesis of the study. Please, summarize the paragraph and re-write it to be clearer and better structured.

We have moved this paragraph to section 3 ahead. It develops the goals of the research, that is true, but it is convenient to place it after the analysis of the coastal retreat. I would say that the reader is brought to the main topic in two phases, first in the introduction, but now more in detail.

**Commented [JP17]:** done

**Commented [JP18]:** done

**Commented [JP19]:** done

[revised manuscript text omitted]

square of flow).

**Commented [JP21]:** done

**Commented [JP22]:** this is not changed; the meaning is different

**Commented [JP23]:** done

**Commented [JP24]:** This is not added to the paper All reservoirs reduce peak flows, that for sure. Our reasoning is not made here for unsteady flow but for steady flow. The effect of flow regulation is treated in turn in the next section.

**Commented [IC25]:** This reasoning is valid for sediment coarser than sand, but this type of sediment is not responsible of delta growth. The effect of reservoirs on sand transport is quite immediate ad it is immediate the growth of the river mouth when big floods carrying a lot of sand occur. Please, change the paragraph accordingly.

I agree with the reviewer. Now, the distinction between gravel and sand is made in the text.

[revised manuscript text omitted]

**9 River engineering: carrying capacity**

A cross-section representative of each date and each reach was drawn with the aid of aerial photographs and archival documents. One example is fig. 7 for reach 1 (see also the sketch in fig.6 for a section in the border between reach 2 and 3).
980 Assuming uniform flow and bed shear stress proportional to hydraulics radius and bed slope (table 3), we have applied the bed

**Commented [IC33]:** Why not discussed here, please, explain.

In the discussion we had to recall to reaches upstream of the lower Llobregat and to periods before the 1946-1981 period of interest at the moment. We are still focusing 1946-1981.

**Commented [IC34]:** Is not a degradation of the fluvial system?

It would have been degradation, definitely, if the downstream reach would have kept its width unchanged.

**Commented [IC35]:** The sediment yield to the delta? Again, the supply that matter to delta growth is sand, and comes from all over the catchment.
Yes, I understand your point of view. Ours is that the Llobregat is a gravel-bed river with a transition to sandy river just at the delta border between reach 3 and 4. This kind of transitions are described in the literature (references are given already in the text), though it may shock that a gravel-bed river suddenly turns into a sandy river. Sand is a not negligible fraction (f.e. 30%) in gravel-bed rivers (and once it is a sandy channel, almost 100%). This sand is therefore transported under the mechanisms of supply and carrying capacity, etc.

**Commented [IC36]:** Don't need general sentences like this.
It has been removed

[revised manuscript text omitted]

---

## Author Response (AR3)

**TRACKED CHANGES**

[revised manuscript text omitted]

 POINT-BY-POINT ANSWER TO REVIEWER

List of answers (in blue), except for the answer "done" to accepted corrections, to the comments (in black).

730 COMMENTS SENT TO AUTHORS. NUMBERING OF LINES ACCORDINGLY

Abstract

The abstract does not reflect properly the content of the paper and is not well structured, it must be re-written.

We made some changes, more in the last version.

735

l.17

Are not all rivers alluvial?

Some rivers run on cohesive material.

740 l.39

It is necessary a section of methods to explain in more detail the calculations, as it is now is no clear enough.

This is done as subsections in the appropriate paragraphs. In the introduction, we prefer not to do that. We opened 2 new methodological subsections, namely 2.1 and 10.1.

745

l.41

Is the word coarse adequated for sand?

Yes, sand is classified in fine, medium and coarse

750 l.43

This sentence is not clear, please explain here the hypothesis, such as: we hypothesize that river channelization has been the main cause of coastal retreat in the Llobregat Delta.

This is more developed at the beginning of section 3. We think there, after having seen the coastal change, is more appropriate.

755

l.73

Why? You should explain this separate analysis and why goes at the end.

In our view, the channelization works of 2004 turned the system into a completely different, artificial mouth. The manuscript is not interested in these modern civil engineering works.

However, the fate of silting at the mouth since 2004 provides at the end a closure of a kind of "sediment budget".
A new phrase announcing this is added in the text.

l.98
Prograded or retreated?
Prograded, actually.

l.101
This is also part of Methods, please make a new section with all methods or subsections with methods in all the sections where there are calculations.
A subsection 2.1 is inserted

l.121
This should go in the previous section, as part of the goals and hypothesis of the study. Please, summarize the paragraph and re-write it to be clearer and better structured.
We have moved this paragraph to section 3 ahead. It develops the goals of the research, that is true, but it is convenient to place it after the analysis of the coastal retreat. I would say that the reader is brought to the main topic in two phases, first in the introduction, but now more in detail.

l.177
the reviewer adds: unless the dam has the capacity to reduce peak flows
This is not added to the paper. All reservoirs reduce peak flows, that for sure. Our reasoning is not made here for unsteady flow but for steady flow. The effect of flow regulation is treated in turn in the next section.

l.179
This reasoning is valid for sediment coarser than sand, but this type of sediment is not responsible of delta growth. The effect of reservoirs on sand transport is quite immediate ad it is immediate the growth of the river mouth when big floods carrying a lot of sand occur. Please, change the paragraph accordingly.
I agree with the reviewer. Now, the distinction between gravel and sand is made in the text.

l.207
What does it means?

The rainfall regime may have changed from 1954 to 2020 due to a climate change (f.e. less rainfall now).

l.226

800  Why they are not analysed together, please explain.

We are still focusing the period 1946-1981 for which the coastal retreat has been shown. In the new version, the floods out of the period 1946-1981 are mentioned separately, to be more coherent.

805  l.242

Not clear what it means

The phrase has been changed.

l.263

810  It seem to me confusing. Depending of what aspect is analysed the time frame is different, it would be important to structure this information, maybe though a table in the section of methods.

This phrase is removed.

815  l.288

Why not discussed here, please, explain.

In the discussion we had to recall to reaches upstream of the lower Llobregat and to periods before the 1946-1981 period of interest at the moment. We are still focusing 1946-1981.

820  l.292

Is not a degradation of the fluvial system?

It would have been degradation, definitely, if the downstream reach would have kept its width unchanged.

825  l.299

The sediment yield to the delta? Again, the supply that matter to delta growth is sand, and comes from all over the catchment.

Yes, I understand your point of view. Ours is that the Llobregat is a gravel-bed river with a transition to sandy river just at the delta border between reach 3 and 4. This kind of transitions

830  are described in the literature (references are given already in the text), though it may shock that a gravel-bed river suddenly turns into a sandy river. Sand is a not negligible fraction (f.e.

30%) in gravel-bed rivers (and once it is a sandy channel, almost 100%). This sand is therefore transported under the mechanisms of supply and carrying capacity, etc.

l.302
Don't need general sentences like this.
It has been removed

COMMENTS UPLOADED TO NHESS. NUMBERING OF LINES ACCORDINGLY

l.299
Again for the next sections try to organize it more clearly in goals, methods, results and discussion. As it is now the manuscript each section is like a mini paper.
It is difficult to bring so many aspects of the problem to a systematized way based on the sequence methods-results-discussion. I have made efforts for that. One change made has been new paragraphs for independent parts of the contents, as well as new clearer titles for them. Another change has been to group all the data in a single paragraph.

l.311
here it looks like there should be two values, one with reservoirs and one without.
yes, the are two (with and without reservoirs) in the table and the ratio is the quotient between them.

l.326
replace by: "but is also affected by river channel deepening"
It is better "depth increase". It is the variable 'Y', water depth, that increases, not the bed that deepens or lowers.

l.335
Is "disturbance analysis" the correct way to name what you are doing here? Is a sediment budget of different river reaches, isn't it?
Right, disturbance means differently in maths. Disturbance is now written with "..."

l.338
Eliminate
this comment is useful to help readers to follow table 6.

l.353

Eliminate

if eliminated the flow of reasoning is broken, in my view.

l.355

Are you referring here only to bed load? Even so, I understand the transfer to the next reach, but why to the next period? You don't know how long it takes. Maybe I am missing something, it's not clear to me.

yes, all refers to bed material load (bed load and supension load coming from the bed)
you are right in your question on why the transfer is to the next period; the answer is written below: "This lapse and step of conveyance have been justified in §5 and §8 with the velocity of the disturbance created by a cut of supply". Now, this phrase is moved closer to the explanation of this transfer.

l.370

Do you mean a decrease from 16.000 to 10.000 m3/yr from the beginning to the end of this period? Please, clarify.

yes, it is a decrease. It is clarified now.

l.371

for the same period?

it wasn't; now it is specified

l.382

which one?

the 2000 flood, it is specified now

l.385

but suspended sediments from large floods do not come only from the river bed, also from river banks and all the river network.

yes, I understand the reviewer's point of view. As above, we stress that the Llobregat is a gravel-bed river which turns into a sandy channel in the transition of reach 3 to reach 4. Reach 4 and 5 produce sand for the delta.

l.399

What do you mean? Please, explain better.

done, it is rephrased.

l.437

Eliminate

we are keeping this phrase, because it is a good summary of the whole section.

l.441

I think this is mostly repeated in previous sections, and mixes results and discussion. Please, skip discussions in previous sections or add discussions in all sections and devote this section to an overall final discussion and conclusions.

this is a summary of the findings so far, which is useful as introduction to a new scope of the discussion.

l.481

Why reforestation may narrow rivers? Is not about reducing land erosion in forested areas?

reducing land erosion implies reducing supply of sediment to the channels and so, typically, narrowing.

l.540

What do you mean? Frequent large floods cannot happen again? I am not sure!

since the authors say that the XIX century hidrology was "anomalous", we should accept that it is not likely that those circumstances will occur again. I put probably instead of virtually.

l.547

I guess this text has to be eliminated.

I plaid for this text as a claim for a shift in the focus of these sciences.